# Pleiotropic mutational effects on function and stability constrain the antigenic evolution of influenza haemagglutinin

Timothy C. Yu [1,2], Caroline Kikawa[1,3,4], Bernadeta Dadonaite [1], Andrea N. Loes[1,5], Janet A. Englund[6,7] & Jesse D. Bloom [1,3,5]✉

The evolution of human influenza virus haemagglutinin (HA) involves simultaneous selection to acquire antigenic mutations that escape population immunity while preserving protein function and stability. Epistasis shapes this evolution, as an antigenic mutation that is deleterious in one genetic background may become tolerated in another. However, the extent to which epistasis can alleviate pleiotropic conflicts between immune escape and protein function/stability is unclear. Here we measure how all amino acid mutations in the HA of a recent human H3N2 influenza strain affect its cell entry function, acid stability and neutralization by human serum antibodies. We find that epistasis has entrenched certain mutations so that reverting to the ancestral amino acid identity in earlier strains is no longer tolerated. Epistasis has also enabled the emergence of antigenic mutations that were detrimental to the cell entry function of HA in earlier strains. However, epistasis appears insufficient to overcome the pleiotropic costs of antigenic mutations that impair the stability of HA, explaining why some mutations that strongly escape human antibodies never fix in nature. Our results refine our understanding of the mutational constraints that shape recent H3N2 influenza evolution: epistasis can enable antigenic change, but pleiotropic effects can restrict its trajectory.

The evolution of human influenza viruses is shaped by selection from population immunity. This immune pressure is especially apparent in the evolution of the viral haemagglutinin (HA) protein, which is the major target of neutralizing antibodies[1,2]. Owing largely to antibody-mediated immune pressure, the HA of human H3N2 influenza virus fixes an average of three to four amino acid substitutions per year[3–7].

However, the antigenic evolution of HA is constrained by its essential role in viral fitness, as it binds to the host cell receptor (sialic acid) and mediates fusion of the viral and cell membranes. Many mutations impair these functions, yet human influenza viruses have demonstrated—both in the laboratory and in nature—the ability to rapidly adapt in the face of these constraints[8–15]. Previous work has revealed that epistatic interactions among mutations in HA play a role in facilitating these adaptations. For example, mutations that impair receptor binding can become tolerated in the presence of other mutations that help restore binding[8–14], while mutations that enhance receptor binding can help buffer the effects of deleterious mutations[15].

Although epistasis among mutations affecting receptor binding is an established mechanism for resolving constraints on HA evolution,

[1]Division of Basic Sciences and Computational Biology Program, Fred Hutchinson Cancer Center, Seattle, WA, USA. [2]Molecular and Cellular Biology Graduate Program, University of Washington, Seattle, WA, USA. [3]Department of Genome Sciences, University of Washington, Seattle, WA, USA. [4]Medical Scientist Training Program, University of Washington, Seattle, WA, USA. [5]Howard Hughes Medical Institute, Seattle, WA, USA. [6]Seattle Children's Research Institute, Seattle, WA, USA. [7]Department of Pediatrics, University of Washington, Seattle, WA, USA. ✉e-mail: jbloom@fredhutch.org

it is less clear how epistasis involving other molecular phenotypes shapes evolution. Many studies have used deep mutational scanning to measure the effects of HA mutations at scale, but these measurements have been typically limited to a single phenotype: cell entry or viral replication in cell culture[9–11,16–20]. One phenotype with poorly understood evolutionary constraints is HA acid stability. As influenza virions are internalized into acidifying endosomes, HA undergoes a pH-triggered destabilization from its metastable prefusion form to a conformation that is primed for mediating membrane fusion[21]. While the structural transitions of this conformational change have been characterized in exquisite detail[22], we lack a complete understanding of how mutations to human H3N2 influenza HA affect acid stability, whether these effects impose pleiotropic costs on antigenic evolution and whether such costs can be alleviated through epistasis.

Here we used pseudovirus deep mutational scanning[23,24] to measure how all amino acid mutations in the HA of a recent human H3N2 influenza strain affect cell entry, acid stability and neutralization by human serum antibodies. By comparing these data to amino acid frequencies at each HA site observed in natural viral evolution, we assessed the extent to which epistasis modulates mutation effects on cell entry and acid stability and whether pleiotropic costs on these phenotypes could be alleviated to enable antigenic mutations to fix. The effects of many mutations on cell entry have changed over time—and epistatic interactions enabled an antigenic mutation that was highly deleterious to cell entry in 2009 to eventually fix in 2022. However, the effects of mutations on acid stability show little evidence of epistasis and several antigenic mutations which pleiotropically reduce acid stability have never fixed. Our results indicate that epistasis plays a central role in driving HA evolution, but its contribution to overcoming pleiotropic costs depends on the underlying molecular phenotype.

## Pseudovirus deep mutational scanning of HA from a recent human H3N2 strain

To measure the effects of mutations to HA on different key molecular phenotypes in high-throughput, we used a recently developed pseudovirus deep mutational scanning approach[23,24]. In brief, we generated genotype–phenotype linked pseudovirus libraries where each virion encodes a mutant HA gene in its genome that matches the HA protein expressed on its surface (Extended Data Fig. 1a,b). Each mutant HA is coupled to a unique nucleotide barcode, forming variants whose phenotypic effects can be measured in a single multiplexed experiment via short-read sequencing of the barcodes (Fig. 1a). These pseudoviruses also express the matched H3N2 neuraminidase (NA) on their surfaces, but this NA is supplied from a separate plasmid and not encoded in the pseudovirus genome. Importantly, these pseudoviruses encode no viral genes other than HA and can only undergo a single round of cell entry. These pseudoviruses are therefore not infectious agents capable of causing disease and so provide a safe way to study HA mutations at biosafety level 2.

We created duplicate libraries in the background of the HA from A/Massachusetts/18/2022 (MA22), which was the H3N2 strain included in the 2024–2025 seasonal influenza vaccine[25]. These libraries were designed to contain every possible amino acid mutation in the HA ectodomain (H3 numbering: 1 to 504) for a total of 504 × 19 = 9,576 mutations. The final libraries contained 64,032 and 70,581 barcoded HA variants which covered 98.7% and 99.0% of all possible mutations, respectively (Extended Data Fig. 1c). Most variants contained a single HA mutation (65%), while others contained zero (15%) or several mutations (20%) (Extended Data Fig. 1d). To extract information from the multiply mutated variants, we used global epistasis models[26,27] to disentangle the effects of individual mutations on measured phenotypes (Methods).

## Mutation effects on HA-mediated cell entry

We quantified the effects of HA mutations on pseudovirus entry into MDCK-SIAT1 cells[28], which express high levels of the α2,6-linked sialic acids preferred by human influenza HAs (Fig. 1a,b and Extended Data Fig. 2 and interactive heatmaps at https://dms-vep.org/Flu_H3_Massachusetts2022_DMS/cell_entry.html). A negative effect indicates that the mutation impairs cell entry, potentially via one or a combination of reasons such as impaired receptor binding, fusion competency, HA folding or HA expression. The measurements of mutation effects were highly correlated between the two independent replicate libraries ($r = 0.95$; Extended Data Fig. 3a). We validated the deep mutational scanning measurements of mutation effects on cell entry for 15 mutations with a range of effects on cell entry using conditionally replicative influenza viruses that lack the PB1 gene[29,30] (Extended Data Fig. 4a,b). The titres of the conditionally replicative influenza virions were highly correlated with the cell entry effects obtained by deep mutational scanning ($r = 0.88$; Extended Data Fig. 4c).

The deep mutational scanning showed that the receptor-binding pocket is heavily constrained overall (Fig. 1c), but that the distribution of cell entry effects varies across its different structural regions (Extended Data Table 1). Some areas, such as the base of the receptor-binding pocket, are highly constrained, while others, such as the 150-loop, are more tolerant of mutations (Fig. 1d and Extended Data Fig. 2). These measurements of the tolerance of different regions to mutations with respect to their effects on cell entry correlate with the variability of different sites in HA during the natural evolution of human H3N2 since 1968 (compare mutation effects and entropy among natural sequences in Fig. 1d). For example, the 150-loop has seen substantial divergence during the natural evolution of H3N2 HA, whereas the base of the receptor-binding pocket remains highly conserved, with only a single substitution at site 195 (Y to F) occurring in the early 2020s.

Many mutations within classically defined antigenic regions (epitopes A–E)[31,32] tend to be well tolerated for the cell entry function of HA (Fig. 1e). However, epitopes that partially overlap with the receptor-binding pocket (epitopes A, B and D) are more constrained. Of these, epitope B is the most constrained yet exhibits the highest variability among natural human H3N2 influenza sequences (Fig. 1e). This discrepancy probably reflects the immunodominance of epitope B with respect to antibody neutralization in the human population, which imposes strong positive selection for mutations at sites within the region[33–35]. Therefore, variability observed in natural sequences depends on both the extent of immune pressure at a site as well as constraints on HA function.

The deep mutational scanning also showed that nearly all mutations to the highly conserved fusion loop at sites 330–350 are strongly deleterious to cell entry (Fig. 1f and Extended Data Fig. 2), consistent with the key role of the region in mediating membrane fusion and also with prior deep mutational scanning on HAs from other viral subtypes[17–20,24,36]. Taken together, we have measured the effects of nearly all mutations to a recent H3N2 HA on cell entry; these measurements help to explain the functional constraints on receptor binding, the general plasticity of antigenic regions and the mutational constraint of the fusion loop.

## Mutation effects on HA acid stability

We next used deep mutational scanning to measure the effects of mutations on the acid stability of HA (Fig. 2a,b and Extended Data Fig. 5 and interactive heatmaps at https://dms-vep.org/Flu_H3_Massachusetts2022_DMS/acid_stability.html). A negative effect indicates that the mutation makes the HA less stable (more susceptible to inactivation at acidic pH). Again, we obtained highly correlated measurements between library replicates ($r = 0.9$; Extended Data Fig. 3b). Note we could only measure effects on stability of mutations that retained at least some minimal cell entry function (Extended Data Fig. 3c). We validated the deep mutational scanning measurements of stability effects for a subset of mutations with varying effects using conditionally replicative influenza virions; the effects measured in the deep mutational scanning concord well with those measured in the validation

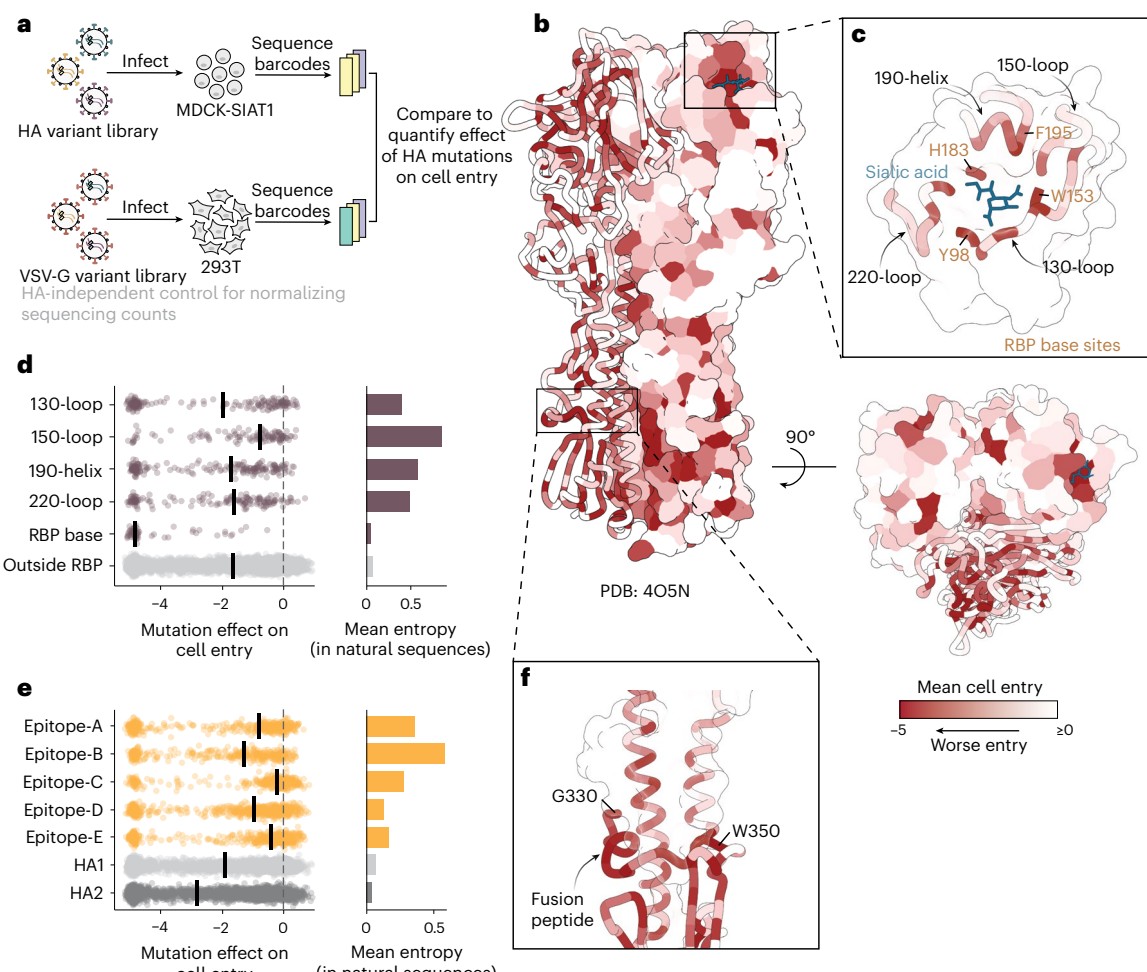

**Fig. 1 | Mutation effects on HA-mediated cell entry. a**, To measure how all HA mutations affect cell entry, we create libraries of barcoded pseudoviruses expressing different HA mutants on their surface (Extended Data Fig. 1). We use deep sequencing to quantify the ability of each HA mutant to enter cells, normalizing the sequencing counts to a copy of the pseudovirus library where all virions express VSV-G and so do not rely on HA for cell entry. The effect of each mutation is quantified as the $\log_2$ of its frequency relative to unmutated variants in the HA condition relative to the VSV-G condition, so negative values indicate impaired cell entry. The mutation effects we report are the median of four measurements, two technical replicates for each of two independently generated

biological replicate libraries (Extended Data Fig. 3a). **b**, Mean effect of mutations at each site on cell entry mapped onto the HA structure (Protein Data Bank 4O5N) viewed from the side or top, with darker red indicating worse cell entry. See Extended Data Fig. 2 for a heatmap of all mutation effects. **c**, Zoomed-in version of the structure showing the receptor-binding pocket (RBP). **d**,**e**, Distribution of mutation effects on cell entry in RBP regions (**d**) and antigenic regions (**e**), with the median effect in each region indicated with a solid black line. The mean Shannon entropy of all sites in each region across a subsampled tree of natural human H3N2 evolution since 1968 is shown on the right. **f**, Zoomed-in version of structure showing sites that make up the fusion peptide and periphery.

assays using influenza virions (Extended Data Fig. 4d). Importantly, the effects of mutations on acid stability are not strongly correlated with the effects of mutations on cell entry, indicating that these assays capture distinct molecular phenotypes (Extended Data Fig. 3d). The phenotypes are distinct for several reasons. First, comparison of natural influenza strains shows that HAs can have acid stabilities that span an appreciable range (for example, fusion pH of 5.0–5.5 in human seasonal strains versus fusion pH of 5.6–6.0 in avian influenza strains[37–39]) but still effectively mediate entry in cells in the laboratory, demonstrating that a range of stabilities are compatible with entry into cell lines even if evolutionary selection for transmissibility in actual human or avian hosts favours a tighter stability range. Second, many mutations that impair cell entry disrupt HA folding, receptor binding or fusion-mediating conformational changes in a manner that is unrelated to acid stability.

Most mutations that affect acid stability destabilize HA (Fig. 2b and Extended Data Fig. 5). The mutations that most strongly affect stability tend to occur in structural regions that participate in the irreversible

HA conformational changes that occur during membrane fusion[22] (Fig. 2c). During this process, the HA1 protomers dilate and eventually dissociate[22,40–42]. Mutations at sites 165, 167, 205 and the 220-loop located in the trimer interface are often destabilizing, probably because they weaken interactions among neighbouring monomers (Fig. 2d). Notably, site 165 carries a high-mannose N-linked glycan that packs against the 220-loop of a neighbouring monomer[43] and all mutations to sites 165 or 167 destabilize, except T167S, the one mutation that preserves the glycan (Fig. 2e). Positively charged arginine residues at sites 220 and 229 are thought to stabilize this region via electrostatic interactions[44] and indeed mutations at these sites destabilize HA (Extended Data Fig. 5).

As HA1 dissociates, the HA1/HA2 interface undergoes conformational changes that are largely driven by intramonomer interactions. Consistent with prior work[42], a tetrad salt bridge involving sites 89, 109, 269 of HA1 and 396 of HA2 is important for HA stability (Fig. 2f and Extended Data Fig. 5). Mutations at site 269 that preserved the net

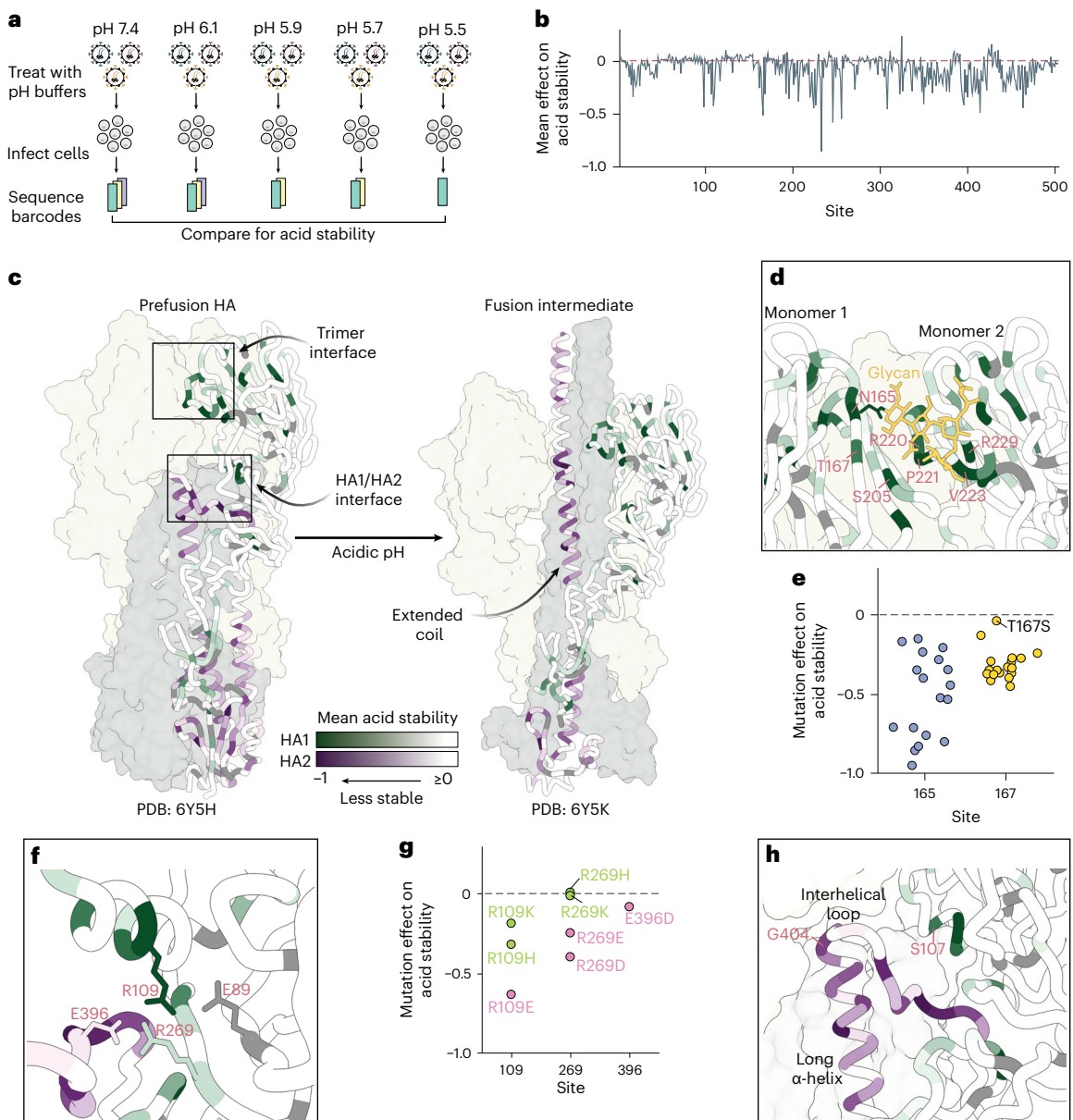

**Fig. 2 | Mutation effects on HA acid stability. a**, We incubated pseudovirus HA variants in different acidic pH buffers before infection and sequenced the pseudovirus barcodes within cells after infection. To quantify mutation effects on acid stability, we compared these barcode counts to those of pseudoviruses treated with neutral pH media, which served as an infection baseline. The mutation effects we report are the median of two biological replicates (Extended Data Fig. 3b). **b**, Mean effect of mutations at each site on acid stability. Negative values indicate sites where mutations decrease stability (for example, lead to viral inactivation at a higher pH). See Extended Data Fig. 5 for a heatmap of all mutation effects. **c**, Mean effect of mutations at each site on acid stability mapped onto the structures of prefusion (Protein Data Bank 6Y5H)

and fusion intermediate (Protein Data Bank 6Y5K) HAs, with darker shades of green and purple indicating greater destabilizing mutation effects in HA1 and HA2, respectively. Dark grey indicates sites where no mutation effects on acid stability were measured as a result of all mutations at these sites strongly impairing cell entry. **d**, Zoomed-in view of the trimer interface, with the same colour scale used in **c**. **e**, Mutation effects on acid stability at sites 165 and 167. **f**, Zoomed-in view of sites that participate in a tetrad salt bridge, with the same colour scale used in **c**. **g**, Mutation effects of charged amino acids at sites that participate in the tetrad salt bridge. **h**, Zoomed-in view of the HA1/HA2 interface, with the same colour scale used in **c**.

charge did not affect stability, but substituting an opposing charged residue was destabilizing (Fig. 2g and Extended Data Fig. 5). Interestingly, charge-preserving mutations to K or H at site R109 were destabilizing, albeit less so than mutation to an oppositely charged E, suggesting that the longer R109 side chain may also play a role in stability.

At acidic pH, the short α-helix, interhelical loop and long α-helix of HA2 form an extended coil[22,45] (Fig. 2c). In the prefusion conformation, these regions feature some of the largest structural differences

between influenza subtypes[46]. The H3/H4/H14 clade-specific G at site 404 in the interhelical loop creates a unique sharp turn that is stabilized in part by a clade-specific S at site 107[46]. As expected, mutations at these sites and in the interacting periphery tend to destabilize HA, indicating the fragile yet essential balance governing this region (Fig. 2h and Extended Data Fig. 5). In summary, we have produced a comprehensive high-throughput map of mutation effects on H3 acid stability; this map provides insight into the structural principles governing H3 HA stability and its conformational transitions.

## Mutations exhibit phenotype-specific entrenchment

Having characterized the effects of mutations to HA on cell entry and acid stability in the background of a recent H3N2 HA, we next explored whether these effects have changed over the last few decades of evolution. To do this, we retraced mutations that swept to fixation in the past (Fig. 3a). During a sweep at a site, both the ancestral amino acid and the descendant amino acid are expected to be functionally tolerated. However, over time the descendant strains may lose tolerance for the ancestral amino acid due to other mutations that become contingent on the current amino acid, a form of epistasis called entrenchment[10,47,48].

The effects of many mutations on cell entry in the receptor-binding pocket have become entrenched over time (Fig. 3b). For instance, a mutation at site 195 from Y to F emerged in 2020 and swept to fixation among human H3N2 strains, but the reversion F195Y in the MA22 background is highly deleterious to cell entry, indicating that the mutation of site 195 from Y to F has become entrenched (Fig. 3b). Y195F was recently shown to be a permissive mutation that enabled the fixation of Y159N and T160I (Extended Data Fig. 6), which confer antigenic benefits and expand receptor specificity in the presence of 195F but impair receptor binding when paired with 195Y[14,16,49]. As the MA22 HA contains 159N and 160I, the reversion F195Y is no longer accessible. Similarly, the G186D mutation emerged in 2020 and subsequently swept to fixation; the reversion D186G in the MA22 background is highly deleterious to cell entry, indicating that G186D has become entrenched (Fig. 3b). This observation is also consistent with recent work that found G186D epistatically interacts with D190N and the pair co-evolved to preserve receptor binding[12]. We validated that conditionally replicative influenza virions encoding MA22 HA with reversions to each of these entrenched mutations (reversions D186G, S193F, F195Y and D225N) had over 100-fold decreased titre relative to virions encoding the unmutated MA22 HA (Extended Data Fig. 4b). Collectively, these results along with previous work[9–12,14], highlight the existence of extensive epistasis with respect to the effects of mutations within the HA receptor-binding pocket on cell entry. This epistasis restricts access to ancestral amino acids and simultaneously opens evolutionary paths for antigenic change (for example, 159N, G186D). However, not all mutations become entrenched, since reversions to ancestral amino acids are observed in HA evolution. Therefore, analysis of entrenchment reveals which reversions are currently accessible or constrained.

In contrast to the extensive epistatic entrenchment involving mutations in the receptor-binding pocket with respect to cell entry, we saw little evidence of entrenchment with respect to cell entry involving mutations in other regions of HA. Nearly all reversions to ancestral amino acids at sites outside the receptor-binding pocket are well tolerated with respect to cell entry with the single exception of T248N (Fig. 3b). Interestingly, the N248T mutation that fixed in the 1980s created an N-linked glycan at N246 that has been maintained ever since. T248S (which is the only mutation that retains this glycan) is noticeably more tolerated than other mutations at sites 246 and 248 (Extended Data Fig. 2), indicating that the glycan is now entrenched. In H3N2 HA evolution, glycosylation near the receptor-binding pocket can shield epitopes from antibodies, but often imposes a fitness cost[50]. Therefore, the N248T mutation—which is located near the receptor-binding pocket—was probably selected by antigenic pressure and mutations that compensated for or became dependent on the glycan led to its entrenchment.

Strikingly, the effects of mutations on acid stability have not become entrenched in any region of HA. Reversions to ancestral amino acids at sites inside and outside the receptor-binding pocket remain well tolerated with respect to acid stability with the single exception of the mildly destabilizing reversion A163V (Fig. 3b). Sites where many mutations destabilize HA tend to have conserved amino acid identities across all natural human H3N2 sequences, which further suggests that

constraints on acid stability may be constant across genetic backgrounds (Extended Data Fig. 7a). At the destabilizing sites that do show variation among natural sequences (for example, sites 219 and 223), the natural mutations are exclusively the particular amino acid changes that do not affect stability (Extended Data Fig. 7b–d). Therefore, it appears that while epistasis commonly shifts the effects of HA mutations on cell entry to entrench mutations, such epistatic processes are much rarer with respect to the phenotype of HA acid stability.

## Epistasis can alleviate the effects of antigenic mutations that impair cell entry but not stability

Phenotypes such as cell entry and acid stability help to determine which HA mutations are tolerated, but immune pressure largely drives positive selection for mutations in HA during the evolution of human influenza viruses. To define this immune pressure on the MA22 HA, we used deep mutational scanning to measure how HA mutations affected neutralization by human sera collected in 2023 from four children born between 2009 and 2021 (Fig. 4a and Extended Data Fig. 8a–d and interactive heatmaps at https://dms-vep.org/Flu_H3_Massachusetts2022_DMS/sera_neutralization.html). We used children's sera since children may play an especially important role in driving influenza evolution[16,51]. The deep mutational scanning measurements are quantified such that a positive effect indicates that the mutation confers viral escape from serum neutralization, whereas a negative effect indicates that the mutation sensitizes the virus to neutralization by the serum. To complement these results, we also used a sequencing-based method[3,52] to measure neutralization titres for each sera against 78 H3N2 strains that either circulated in humans or were included in vaccines between 2012 and 2023.

Our deep mutational scan revealed that mutations at site K140 in the MA22 HA increase sensitivity to neutralization (Fig. 4a, note the negative escape at site 140). Site 140 is in epitope A of HA and changed from an I to K in 2022 (Fig. 4b). The fact that many mutations to site 140, including the reversion K140I, are sensitizing (Extended Data Fig. 8c) suggests that sera contain neutralizing antibodies targeting the ancestral amino acid identity of I140 that were escaped by the I140K mutation in 2022; the MA22 reversion K140I restores neutralization by these antibodies. We validated that conditionally replicative influenza virions with the MA22 HA carrying a K140I reversion increased sensitivity to neutralization in two of the three sera predicted to contain 140I-specific antibodies by the deep mutational scanning (Fig. 4c and Extended Data Fig. 9a,b). For those sera, neutralization of historic and vaccine strains differing by up to 34 amino acids from MA22 could also be explained by site 140, although we cannot rule out the influence of other historical mutations (Extended Data Fig. 9c). Taken together, these results indicate that 140I-specific antibodies contribute substantially to the neutralizing antibody activity of sera from some individuals and suggest that the I140K mutation that fixed in 2022 was probably selected by this immune pressure.

Given that changing site 140 from I to K causes such an appreciable reduction in antibody neutralization for some sera, why did this mutation not spread widely in human H3N2 influenza until 2022 (Fig. 4b)? At least part of the answer appears to be that the I140K mutation was highly deleterious for the cell entry function of HAs from older strains: for instance, I140K causes a nearly 40-fold drop in the titre of conditionally replicative influenza virions encoding the HA from the older A/Perth/16/2009 (H3N2) strain, whereas interchanging I and K at site 140 in the MA22 HA has little impact on the titre (Fig. 4d). These observations suggest that I140K was evolutionarily inaccessible in 2009 because of its strongly negative effect on cell entry function of HA and could not be positively selected until a permissive HA genetic background emerged. This finding parallels the recent demonstration by Lei et al. [12] that G186D and D190N, two mutations that co-evolved together in 2020, are individually deleterious to receptor binding but together restore this function and lead to escape from human sera. Collectively,

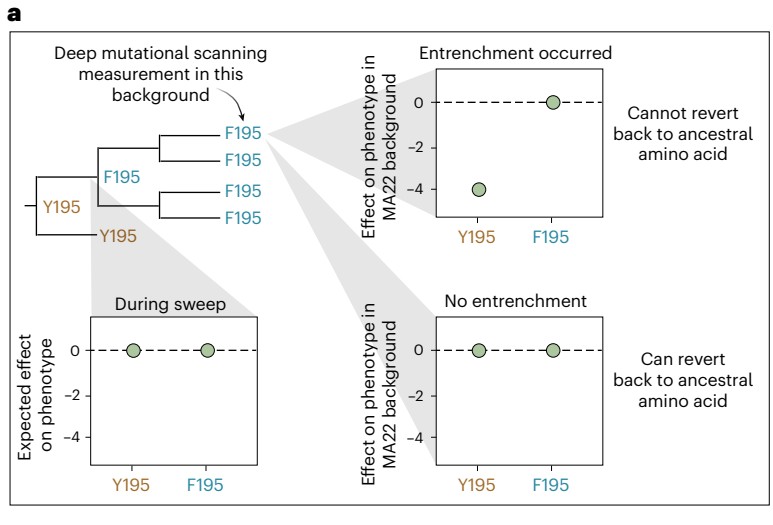

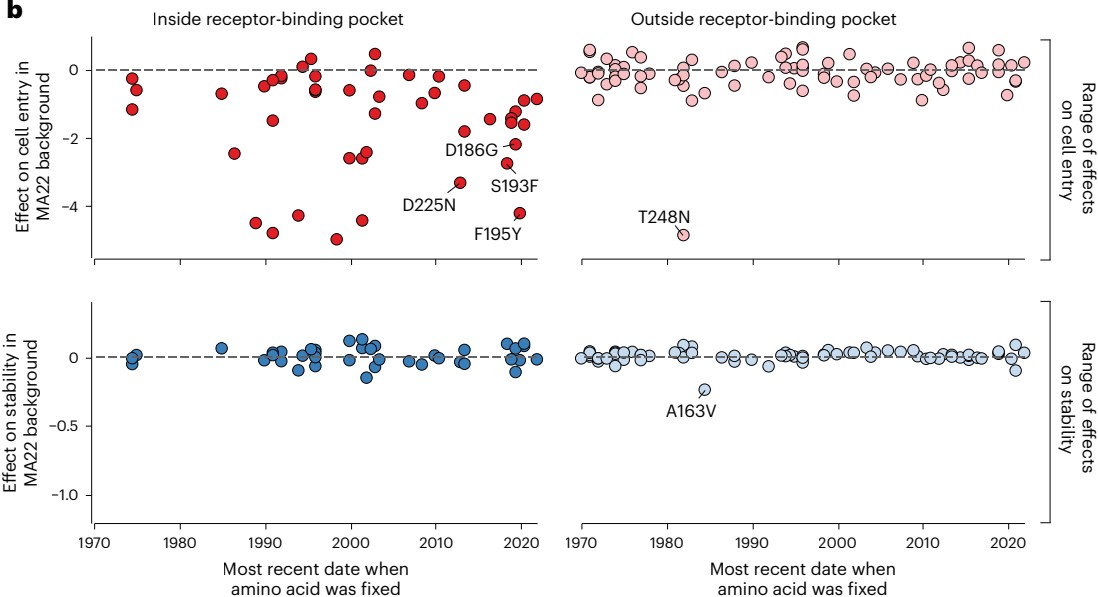

**Fig. 3 | Mutations exhibit phenotype-specific epistatic entrenchment.**
**a**, Hypothetical data illustrating entrenchment of a mutation via epistasis. The phylogenetic tree shows a sweep of F replacing Y at site 195. During the sweep, both F and Y are expected to be tolerated. However, the effect of reverting to the ancestral amino acid (Y) may change over time due to entrenchment. The HA used for deep mutational scanning contains the current amino acid (F) at site 195. If entrenchment occurred, then reverting to the ancestral amino acid Y195 will be deleterious in the recent genetic background used for deep mutational scanning. In the absence of entrenchment, Y195 remains tolerated in newer genetic backgrounds. **b**, Actual experimental data showing the effects of reversions to all ancestral amino acids that were previously fixed in human H3N2 strains since 1968 on HA-mediated cell entry (red, top row) or HA acid stability

(blue, bottom row) as measured in the deep mutational scanning on the MA22 HA. Mutations are placed on the *x* axis by the most recent date that the ancestral amino acid was fixed and the panel columns indicate whether the mutation is at a site inside (left column) or outside (right column) the receptor-binding pocket. The range of the *y* axis for each phenotype is set to span the range of effects of all mutations to HA (not just those that fixed during natural H3N2 evolution) in the deep mutational scanning. There is extensive epistatic entrenchment of mutations in the receptor-binding pocket with respect to cell entry, but no substantial entrenchment of mutations with respect to stability. To mouseover the individual mutations, see https://dms-vep.org/Flu_H3_Massachusetts2022_DMS/entrenchment.html for an interactive version of this plot.

these results indicate that epistasis can alleviate pleiotropic constraints on cell entry function and facilitate antigenic evolution.

We next examined whether we could identify pleiotropic constraints on other mutations that affect serum neutralization of the MA22 HA. Mutations at a variety of HA sites reduce serum neutralization, including sites 145, 165, 189, 205, 220 and 229 (Fig. 4a, note positive escape at these sites). In the absence of other constraints, one might expect recent evolution to select for neutralization escape mutations at these sites. In some cases, this is occurring: S145N and K189R are single nucleotide accessible mutations from MA22 that cause sera escape in the deep mutational scanning (Fig. 4e and Extended Data Fig. 9d) and

both have been increasing in frequency among human H3N2 sequences since 2024 (Fig. 4b). However, some of the sites that strongly escape sera neutralization show no variation over decades of H3N2 HA evolution. For instance, mutations at sites 165, 205, 220 and 229 cause strong serum escape in the deep mutational scanning (Fig. 4a,e) and in validation assays with conditionally replicative influenza virions (Fig. 4f), but these sites have not changed during natural HA evolution (Fig. 4b). This disparity between the abundance of single nucleotide accessible escape mutations identified by deep mutational scanning and their absence in natural human H3N2 sequences suggests that pleiotropic constraints could be constraining evolution at these sites.

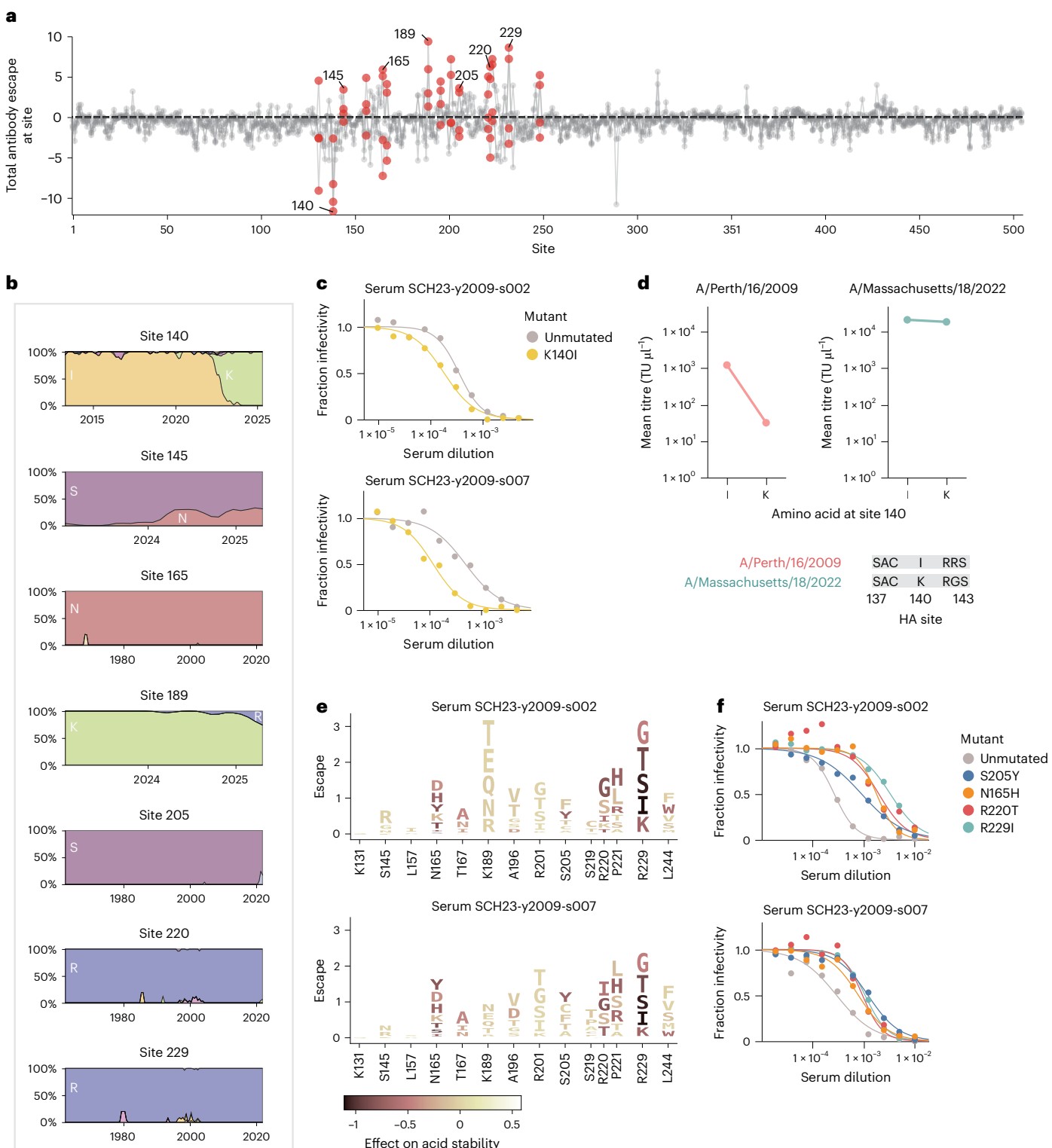

**Fig. 4 | During H3N2 evolution, epistasis alleviates the effects of antigenic mutations that impair cell entry but not stability. a**, Sum of mutation effects at each site in the MA22 HA on escape from four human sera collected in 2023. Key sites of escape or sensitization included in logoplots are coloured red and sites where escape mutants were validated in independent neutralization assays are labelled with text. See Extended Data Fig. 8 for data for individual sera. **b**, Frequencies of amino acids observed in natural human H3N2 HA sequences over time (x axis indicates year) for several key sites. Note the x axis range varies as some subplots show changes in recent evolution, while others show conservation across decades of evolution. **c**, Neutralization by two human sera of conditionally replicative influenza virions with MA22 HA with or without the mutation K140I. Each point is the mean of two technical replicates. **d**, Titre of

conditionally replicative influenza virions with HA from either A/Perth/16/2009 or MA22 with either an I or a K amino acid at site 140. Each point is the mean of four titre measurements, two technical replicates from the same virion rescue stock and two biological replicate stocks rescued from independent plasmid preparations. **e**, Logoplots showing amino acid mutations to the MA22 HA that are accessible by single nucleotide changes. The height of each letter is proportional to the escape from the indicated serum as measured by deep mutational scanning. Each mutation is coloured by its effect on HA acid stability as measured in the deep mutational scanning, with darker colours indicating decreased stability. **f**, Neutralization of conditionally replicative MA22 virus with destabilizing mutations by the two sera in **e**. Each point is the mean of two technical replicates.

Notably, many of the escape mutations at these sites destabilize HA in our deep mutational scanning (Fig. 4e), even if they are not directly deleterious for HA-mediated cell entry. For instance, the serum escape mutations N165H, S205Y, R220T and R229I are all roughly neutral with respect to HA-mediated cell entry (Extended Data Fig. 4b), yet mutations at these sites strongly destabilize HA, making it more sensitive to acid inactivation (Extended Data Fig. 4d) while conferring escape from sera in independent neutralization assays. Therefore, the effects of mutations on the acid stability of HA may impose a strong pleiotropic constraint on its evolution, even when these mutations have no apparent effect on HA-mediated cell entry.

## Discussion

The extent to which pleiotropic conflicts constrain evolution of H3N2 HA has remained unclear. Here we measured the effects of all amino acid mutations to a recent H3 HA on cell entry, acid stability and neutralization by serum antibodies. Interpreting these effects in the context of the natural evolution of HA reveals that recent H3N2 evolution has alleviated constraints on cell entry through epistasis, but some antigenic mutations with no effect on cell entry remain highly constrained because they pleiotropically decrease acid stability.

Other studies have reported epistatic entrenchment to be common with respect to the impact of HA mutations within the receptor-binding pocket on viral replication[9–12,14]. Our work also finds that HA mutations in the receptor-binding pocket have become entrenched with respect to their effects on cell entry, but this pattern is observed much less often for HA mutations outside the receptor-binding pocket. Furthermore, epistatic entrenchment with respect to acid stability is absent across the entire HA during the timeframe we analysed, demonstrating how epistasis can be phenotype-specific. Why might mutation effects on acid stability be less prone to shift due to epistasis? A variety of studies have found that mutations to proteins often have roughly additive effects on stability[53,54] and epistasis at the level of function tends to arise from the nonlinear relationship between stability and function rather than underlying epistasis in the effects of mutations on stability[55,56]. In contrast, the receptor-binding pocket involves a network of amino acid residues positioned so that their side chains interact with sialic acid via hydrogen bonds and other non-covalent contacts. Mutations at one site can alter these interactions in ways that modify the effects of changes at interacting sites[9–14], providing a structural basis for extensive epistasis with respect to cell entry. In a more abstract view, acid stability could represent a single underlying 'global' biophysical property that is less influenced by epistasis[26,57], whereas cell entry is a higher-order phenotype that involves several underlying properties (for example, receptor binding, protein stability and membrane fusion). There is evidence for other proteins that mutations often have additive effects on underlying biophysical properties and epistasis in higher-order phenotypes often arises simply from their nonlinear dependence on underlying molecular properties[26,58–60].

Our deep mutational scanning shows that there are sites outside classically defined antigenic regions where mutations strongly escape serum antibody neutralization but also destabilize HA (for example, sites 165, 205, 220 and 229). Interestingly, these sites are highly conserved and located in the trimer interface. While non-neutralizing anti-H3 and neutralizing anti-H7 antibodies targeting this region have been described[61–64], to our knowledge, humans have not been shown to possess neutralizing anti-H3 trimer interface antibodies. While it is unclear how these mutations reduce serum antibody neutralization, we speculate that there are two possible mechanisms. These mutations could abrogate binding of antibodies that directly target the trimer interface, nearby epitopes or across protomers[65]. Alternatively, the destabilizing effect could accelerate membrane fusion, a previously reported strategy for antibody escape[66]. In any case, our work suggests that antibodies targeting these sites may be particularly difficult to escape due to strong pleiotropic constraints on destabilizing mutations during HA evolution.

Overall, our study highlights how mutations to HA often have pleiotropic effects. The extent to which pleiotropic constraints can be alleviated by epistasis differs across phenotypes: for example, constraints on cell entry appear more readily alleviated than constraints on acid stability in recent H3N2 influenza HA evolution. Whether these constraints similarly shape the evolution of other H3N2 strains and influenza subtypes remains to be determined. A deeper understanding of how pleiotropy and epistasis shape HA evolution will be useful for forecasting viral evolution and designing therapeutics that are resistant to viral escape.

## Methods

### Cell lines and media

The following cell lines were used: 293T (ATCC, CRL-3216), 293T-rtTA (from ref. 23), 293T-CMV-PB1 (from ref. 67), MDCK-SIAT1 (HPA Cultures, 05071502), MDCK-SIAT1-CMV-PB1 (from ref. 67) and MDCK-SIAT1-CMV-PB1-TMPRSS2 (from ref. 19).

All cell lines were maintained in D10 media (Dulbecco's Modified Eagle Medium supplemented with 10% heat-inactivated fetal bovine serum, 2 mM L-glutamine, 100 U ml⁻¹ of penicillin and 100 µg ml⁻¹ of streptomycin). To suppress rtTA activation, 293T-rtTA cells were grown in tetracycline-free D10, which is made with tetracycline-negative fetal bovine serum (Gemini Bio, ref. no. 100-800) instead. For virus rescue and infection with HA expressing pseudovirus libraries and conditionally replicative influenza viruses, we used influenza growth media (IGM, Opti-MEM supplemented with 0.01% heat-inactivated fetal bovine serum, 0.3% bovine serum albumin, 100 µg ml⁻¹ of calcium chloride, 100 U ml⁻¹ of penicillin and 100 µg ml⁻¹ of streptomycin) or neutralization assay media (NAM, medium-199 supplemented with 0.01% heat-inactivated fetal bovine serum, 0.3% bovine serum albumin, 100 µg ml⁻¹ of calcium chloride, 100 U ml⁻¹ of penicillin, 100 µg ml⁻¹ of streptomycin and 25 mM HEPES).

### Plasmids and primers

Plasmid maps are available via GitHub at https://github.com/dms-vep/Flu_H3_Massachusetts2022_DMS/tree/main/data/supplemental_data/plasmids.

Primer sequences are available via GitHub at https://github.com/dms-vep/Flu_H3_Massachusetts2022_DMS/tree/main/data/supplemental_data/primers.

### Human sera

Human sera samples were obtained from Seattle Children's Hospital during routine blood draws from children receiving medical care in December 2023. This was approved by the Seattle Children's Hospital Institutional Review Board with a waiver of consent. Sera samples were treated with receptor-destroying enzyme (RDE) and heat-inactivated to remove non-specific inhibitors before use in deep mutational scanning library selections and neutralization assays. RDE was prepared by resuspending one vial of lyophilized RDE II (Seikan) in 20 ml of PBS and filtering through a 0.22-µM filter. Sera and RDE were combined at a 1:3 sera to RDE ratio, incubated at 37 °C for 2.5 h and then incubated at 55 °C for 30 min (ref. 68). RDE-treated sera were stored at −80 °C until further use.

### Design of deep mutational scanning libraries

We used a lentiviral backbone that is schematized in Extended Data Fig. 1a[23]. The libraries were designed in the background of the A/Massachusetts/18/2022 HA, which was the 2024–2025 cell-based vaccine strain. The HA gene was codon-optimized via the GenSmart codon optimization tool offered by GenScript, as we found this codon optimization increases viral titre. The plasmid map for the lentiviral backbone with codon-optimized HA sequence is available via GitHub at

https://github.com/dms-vep/Flu_H3_Massachusetts2022_DMS/blob/main/data/supplemental_data/plasmids/4570_pH2rU3_ForInd_Massachusetts2022HA_GenscriptV1_T7_CMV_ZsGT2APurR.gb. We aimed to include all single amino acid mutations in the HA ectodomain (H3 numbering 1–504). Twenty stop codons located at alternating positions from the start of the ectodomain were also included as negative controls for cell entry measurements. We ordered a site-saturation variant library with these specifications from Twist Biosciences. The final Twist quality control report for the library is available via GitHub at https://github.com/dms-vep/Flu_H3_Massachusetts2022_DMS/blob/main/data/supplemental_data/Final_QC_Twist_VariantProportion.csv.

### Cloning of deep mutational scanning plasmid library

We cloned the deep mutational scanning plasmid libraries following an approach first described in ref. 23. Barcoding PCR was performed using the Twist library as template to append random 16 nucleotide barcodes downstream of the HA gene stop codon. A total of 5 ng of Twist library (1 µl) was combined with 1.5 µl of ForInd_AddBC_2 primer (10 µM), 1.5 µl of 5′for_lib_bcing primer (10 µM), 21 µl of molecular biology grade water and 25 µl of KOD hot start master mix (ThermoFisher, ref. no. 71842-4). The PCR cycling conditions were:

(1) 95 °C, 2 min
(2) 95 °C, 20 s
(3) 55.5 °C, 20 s, cooling at 0.5 °C s$^{-1}$
(4) 70 °C, 1 min
(5) Return to step 2, 9 cycles
(6) 12 °C hold

The barcoding was performed in two independent reactions, yielding two barcoded PCR products to serve as biological library replicates (libraries A and B). Therefore, the two libraries contain unique barcodes and all subsequent cloning and virus generation steps were carried out separately for each library.

The lentiviral backbone was digested from a plasmid containing mCherry in place of the HA insert (3137_pH2rU3_ForInd_mCherry_CMV_ZsGT2APurR) by incubating with XbaI and MluI for 2 h at 37 °C, followed by 20 min at 65 °C to inactivate XbaI.

Both the barcoded HA libraries and digested lentiviral backbone were run on a 0.8% agarose gel and bands of the expected size were excised and purified using the NucleoSpin Gel and PCR Clean-up kit (Macherey-Nagel, catalogue no. 740609.5) followed by additional purification with Ampure XP beads (Beckman Coulter, catalogue no. A63881) to ensure high purity. All were eluted in molecular biology grade water.

Barcoded HA libraries were cloned into the lentiviral backbone at a 1:2 insert to vector ratio in a 1-h Hifi assembly reaction using the NEBuilder HiFi DNA Assembly kit (NEB E5520S). The Hifi reactions were purified with Ampure XP beads and eluted in molecular biology grade water, then transformed into 10-beta electrocompetent cells (NEB, catalogue no. C3020K) using a BioRad MicroPulser Electroporator (catalogue no. 1652100), shocking at 2 kV for 5 ms. Fifteen electroporation reactions were performed for each library and bacteria were plated on 15-cm LB + ampicillin plates and grown overnight at 37 °C. The next day, colonies were scraped with LB + ampicillin and plasmids were extracted using the QIAGEN HiSpeed Plasmid Maxi Kit (catalogue no. 12662). The total number of colonies for Library A and B were 8.4 × 10$^6$ c.f.u. and 7.5 × 10$^6$ c.f.u., respectively. Large numbers of colonies at this stage are necessary to ensure that library diversity does not become bottlenecked.

On the basis of the Twist quality control report, 35 sites were missing >75% of mutations and 19 mutations at other sites were missing in the library. Therefore, we aimed to clone a 'spike-in' plasmid library that contains these missing mutations using a mutagenesis PCR protocol[24,69]. We designed NNS primers for missing sites with https://github.com/jbloomlab/CodonTilingPrimers and primers for missing mutations with https://github.com/jbloomlab/TargetedTilingPrimers. Forward and reverse primer pools were created by combining either forward or reverse NNS and targeted mutation primers at an equal molar ratio per codon. To prepare a linear HA template for mutagenesis PCR, the lentiviral backbone plasmid encoding the codon-optimized HA was incubated with NotI and NdeI for 2 h at 37 °C, followed by 20 min at 65 °C to inactivate both enzymes. The digest product was run on a 0.8% agarose gel and the band corresponding to the linear HA fragment was purified.

The protocol for mutagenesis involves two reactions: a mutagenesis PCR and a joining PCR. Two separate replicates were performed to form biological library replicates (spike-in libraries A and B) as was done for the Twist libraries. The first mutagenesis PCR was divided into forward and reverse reactions. Both forward and reverse reactions shared the following PCR conditions: 4 µl of linear HA template (3 ng µl$^{-1}$), 8 µl of molecular biology grade water and 15 µl of KOD hot start master mix. A total of 1.5 µl of the forward primer pool (5 µM) and 1.5 µl of 3′rev_linjoin_KHDC primer (5 µM) were added to the forward reaction. A total of 1.5 µl of the reverse primer pool (5 µM) and 1.5 µl of VEP_Amp_For primer (5 µM) were added to the reverse reaction. The PCR cycling conditions were:

(1) 95 °C, 2 min
(2) 95 °C, 20 s
(3) 70 °C, 1 s
(4) 54 °C, 20 s, cooling at 0.5 °C s$^{-1}$
(5) 70 °C, 50 s
(6) Return to step 2, 9 cycles
(7) 4 °C hold

The forward and reverse mutagenesis PCR products were diluted 1:4 in molecular biology grade water and 4 µl of each were added to the joining PCR reaction along with the following: 1.5 µl of 3′rev_linjoin_KHDC primer (5 µM), 1.5 µl of VEP_Amp_For primer (5 µM), 4 µl of molecular biology grade water and 15 µl of KOD hot start master mix. The PCR cycling conditions were:

(1) 95 °C, 2 min
(2) 95 °C, 20 s
(3) 70 °C, 1 s
(4) 54 °C, 20 s, cooling at 0.5°C s$^{-1}$
(5) 70 °C, 65 s
(6) Return to step 2, 19 cycles
(7) 4 °C hold

A DpnI digest was performed afterwards to remove any potential unmutated HA template (which would be methylated) by incubating the joining PCR products with DpnI for 20 min at 37 °C. DpnI-digested joining PCR products were run on a 0.8% gel and the expected bands were excised and purified as described above. The mutagenized HA fragments were barcoded and cloned into the lentiviral backbone following the same approach described above for the Twist library. However, spike-in plasmids were extracted using the QIAprep Spin Miniprep Kit (Qiagen, catalogue no. 27104) instead of via maxipreps.

Corresponding replicates of the Twist plasmid libraries and spike-in plasmid libraries were combined at a 1:4 Twist to spike-in molar ratio per codon, with the spike-in library intentionally added at four times the amount required for an equal per-codon molar ratio because long-read PacBio sequencing of the plasmid libraries revealed that this ratio results in the most even distribution of mutants in the combined libraries.

### Production of cell-stored deep mutational scanning libraries

Deep mutational scanning requires genotype–phenotype linked pseudoviruses. The rationale for this is described in detail in the caption of Extended Data Fig. 1b. We generated cell-stored deep mutational scanning libraries where each cell is integrated with a single copy of a

barcoded HA mutant to enable rescue of genotype–phenotype linked pseudoviruses[23,24]. Some 15-cm plates were plated with ~20 million 293T cells. On the next day, each plate was transfected with 12.5 µg of plasmid library encoding the lentiviral backbone with barcoded HA mutants, 3.125 µg of each lentiviral helper plasmid (26_HDM_Hgpm2, 27_HDM_tat1b and 28_pRC_CMV_Rev1b), 3.125 µg of plasmid expressing vesicular stomatitis virus glycoprotein (VSV-G) (29_HDM_VSV_G) and 3.125 µg of plasmid expressing a strain-matched codon-optimized NA gene (4576_HDM_Massachusetts2022NA_Genscript). BioT transfection reagent (Bioland Scientific, catalogue no. B01-02) was used according to manufacturer's instructions. Note the NA is important because the virions produced here will also have HA expressed on their surface from the lentiviral backbone. The NA expression prevents HA from binding to producing cells, as this binding could bias the library since the HA mutants will have different abilities to bind to sialic acids on the producing 293T cells. At 48 h after transfection, the supernatant was filtered through a 0.45-µm syringe filter (Corning, catalogue no. 431220) and stored at −80 °C. Then 4 × 15-cm plates were transfected with each library replicate, resulting in ~100 ml of VSV-G pseudotyped library viruses. An aliquot of these viruses was used to infect 293T cells and the titre in transcription units (TU) per millilitre was determined by measuring the percentage of zsGreen positive cells via flow cytometry.

The VSV-G pseudotyped library viruses were used to infect 293T-rtTA cells at a multiplicity of infection (MOI) of 0.7% to ensure that each cell integrates at most one copy of provirus. The MOI was confirmed by measuring the percentage of zsGreen positive cells via flow cytometry at 48 h after transduction. On the basis of the measured MOI and number of cells present during infection, cells were pooled such that each library would contain an estimated 60,000 infected cells. This number was chosen to be high enough to ensure each mutant is associated with multiple barcodes to increase measurement accuracy (for ~10,000 mutants, each mutant would have about six barcodes), while also being low enough that it would be possible to measure all variants given our pseudovirus titre in selection experiments. Note the final numbers ended up being close to this target: 64,032 for library A and 70,581 for library B (Extended Data Fig. 1c). Integrated cells were selected by growing in the presence of 0.75 µg ml⁻¹ of puromycin for 1 week (fresh media with puromycin was replenished every 48 h). After selection was complete, integrated cells were expanded in tetracycline-free D10 for 24 h and then frozen down in liquid nitrogen in 2 × 10⁷-cell aliquots for long-term storage.

### Rescue of HA and VSV-G expressing pseudovirus libraries

To rescue HA expressing pseudoviruses from the integrated cells, 150 million cells were plated in five-layer flasks in tetracycline-free D10 supplemented with 1 µg ml⁻¹ of doxycycline to induce HA expression from the integrated genomes. On the next day, each flask was transfected with 43.75 µg of each helper plasmid (26_HDM_Hgpm2, 27_HDM_tat1b and 28_pRC_CMV_Rev1b), 15 µg of plasmid expressing human airway trypsin-like protease (3781_HDM_HAT) to activate HA for membrane fusion and 3.75 µg of plasmid expressing NA (4576_HDM_Massachusetts2022NA_Genscript). BioT transfection reagent was used according to manufacturer's instructions. At 16 h after transfection, the tetracycline-free D10 in each flask was aspirated and 150 ml of IGM supplemented with 1 µg ml⁻¹ of doxycycline was added. This swap to low-serum media is absolutely necessary because non-specific inhibitors in FBS can inactivate HA and interfere with pseudovirus infection. At 32 h after media swap, the supernatant was filtered through a 0.45-µm SFCA Nalgene 500-ml Rapid-Flow filter unit (catalogue no. 09-740-44B). Filtered supernatant was then concentrated by adding LentiX Concentrator (Takara, catalogue no. 631232) at a 1:3 virus to concentrator ratio, incubating at 4 °C overnight and spinning at 1,500g and 4 °C for 45 min. Following centrifugation, supernatant was discarded and viral pellets resuspended

in NAM to an estimated titre of ~2 × 10⁶ TU ml⁻¹. Then 1-ml aliquots of concentrated HA expressing pseudoviruses were frozen at −80 °C for use in downstream selection experiments.

To rescue VSV-G expressing pseudoviruses from integrated cells, 30 million cells were plated in 10-cm plates in tetracycline-free D10. On the next day, each plate was transfected with 7.3125 µg of each helper plasmid (26_HDM_Hgpm2, 27_HDM_tat1b and 28_pRC_CMV_Rev1b), 0.75 µg of plasmid expressing NA (4576_HDM_Massachusetts2022NA_Genscript) and 7.3125 µg of plasmid expressing VSV-G (29_HDM_VSV_G). At 48 h after transfection, the supernatant was filtered through a 0.45-µm SFCA Nalgene 500-ml Rapid-Flow filter unit and concentrated using LentiX Concentrator but viral pellets were resuspended in D10. Aliquots of concentrated VSV-G expressing pseudoviruses were frozen at −80 °C for use in linking mutations to barcodes and cell entry selection experiments.

### Long-read sequencing to link mutations to barcodes

A total of 1 × 10⁶ 293T cells were plated in each well of six-well plates coated with poly-L-lysine to help with cell adhesion. On the next day, 15 million TU of VSV-G expressing pseudoviruses that were rescued from cell-stored deep mutational scanning libraries were used to infect the cells. At 12 h after infection, the non-integrated reverse-transcribed lentiviral genomes were recovered by miniprepping the 293T cells using the QIAprep Spin Miniprep Kit.

Amplicons for long-read sequencing of the miniprepped genomes were prepared by following an approach described in ref. 23. Briefly, the eluted minipreps were split into two separate reactions so that each could be uniquely tagged for detecting strand exchange events from the PCR. The number of PCR cycles was chosen intentionally to limit the possibility of strand exchange. Both reactions shared the following PCR conditions: 20 µl of KOD hot start master mix and 18 µl of miniprepped DNA; 1 µl of 5_PacBio_G primer (10 µM) and 1 µl of 3_PacBio_C primer (10 µM) were added to the first reaction; 1 µl of 5_PacBio_C primer (10 µM) and 1 µl of 3_PacBio_G primer (10 µM) were added to the second reaction. The PCR cycling conditions were:

(1) 95 °C, 2 min
(2) 95 °C, 20 s
(3) 70 °C, 1 s
(4) 60 °C, 10 s, cooling at 0.5 °C s⁻¹
(5) 70 °C, 60 s
(6) Return to step 2, 7 cycles
(7) 70 °C, 1 min
(8) 4 °C hold

The round 1 PCR products were purified with 50 µl of Ampure XP beads and eluted in 35 µl of elution buffer. For each library, equal volumes of the two separate round 1 PCR reactions were pooled. The round 2 PCR reactions contained: 25 µl of KOD hot start master mix, 21 µl of pooled round 1 product, 2 µl of 5_PacBio_Rnd2 primer (10 µM) and 2 µl of 3_PacBio_Rnd2 primer (10 µM). The PCR cycling conditions were:

(1) 95 °C, 2 min
(2) 95 °C, 20 s
(3) 70 °C, 1 s
(4) 60 °C, 10 s, cooling at 0.5 °C s⁻¹
(5) 70 °C, 1 min
(6) Return to step 2, 10 cycles
(7) 70 °C, 1 min
(8) 4 °C hold

The round 2 PCR products were purified with 50 µl of Ampure XP beads and eluted in 40 µl of elution buffer. PCR reactions for each library were combined and amplicon length was verified by TapeStation before sequencing. Libraries were sequenced on a single SMRT cell with a movie length of 30 h on a PacBio Sequel IIe sequencer. For details on

computational analysis, see the section below on 'PacBio sequencing analysis'.

## Mutation effects on cell entry

To measure effects of HA mutations on cell entry, we followed the approach described in ref. 23. Briefly, we infected MDCK-SIAT1 cells with the HA expressing pseudovirus library and infected 293T cells with the VSV-G expressing pseudovirus library. The VSV-G expressing library is necessary to provide a baseline for infection as VSV-G can mediate cell entry without relying on HA. We used 293T cells for VSV-G infection because titre of VSV-G expressing pseudoviruses are higher when infecting 293T cells compared with MDCK-SIAT1 cells.

A total of $1 \times 10^6$ 293T cells in D10 or $7 \times 10^5$ MDCK-SIAT1 cells in NAM were plated in each well of six-well plates. A total of 2.5 µg ml$^{-1}$ of amphotericin B was added to the MDCK-SIAT1 cells when plating as this improves HA pseudovirus titre. On the next day, we infected the MDCK-SIAT1 cells with ~$1.2 \times 10^6$ TU of HA pseudovirus library and the 293T cells with ~$8 \times 10^6$ TU of VSV-G pseudovirus library. Before infection, the HA pseudovirus library was treated with 500 nM of oseltamivir for 20 min on ice to inhibit NA from interfering with cell entry. Note infections with the HA pseudovirus library must be done in NAM, as the serum in D10 contains non-specific inhibitors that inhibit H3 infection.

At 12 h after infection, the non-integrated reverse-transcribed lentiviral genomes were recovered by miniprepping the 293T and MDCK-SIAT1 cells. To prepare the amplicons for Illumina sequencing, two rounds of PCR were performed: the first round appends the Illumina Truseq read 1 and read 2 sequences and the second round attaches indices for multiplexing. The round 1 PCR reactions contained: 22 µl of miniprepped DNA, 25 µl of KOD hot start master mix, 1.5 µl of Illumina_Rnd1_For primer (10 µM) and 1.5 µl of Illumina_Rnd1_Rev3 primer (10 µM). The PCR cycling conditions were:

(1) 95 °C, 2 min
(2) 95 °C, 20 s
(3) 70 °C, 1 s
(4) 58 °C, 10 s, cooling at 0.5 °C s$^{-1}$
(5) 70 °C, 20 s
(6) Return to step 2, 27 cycles
(7) 70 °C, 1 min
(8) 4 °C hold

Round 1 PCR products were purified with 150 µl of Ampure XP beads and eluted in 50 µl of elution buffer. Concentrations of each PCR product were determined by Qubit 4 Fluorometer (ThermoFisher, ref. no. Q33238). The round 2 PCR reactions contained: 20 ng of round 1 PCR product, 25 µl of KOD hot start master mix, 2 µl of each of the round 2 indexing primers (10 µM each) and up to 25 µl of molecular biology grade water. The same PCR cycling conditions as round 1 were used, except only 20 cycles were performed. Concentrations of each round 2 PCR product were determined by Qubit 4 Fluorometer. The samples were then pooled in equal DNA amounts and run on a 1% agarose gel. The correct size band (283 base pairs) was excised, purified with Ampure XP beads, diluted to a concentration of 4 nM and sequenced on an Illumina NextSeq 2000 (with P2 reagent kit) or NovaSeq X Plus system. For details on how sequencing counts were converted to mutation effects on cell entry, see the section below on 'Illumina sequencing barcode analysis'.

## Mutation effects on acid stability

To measure effects of HA mutations on acid stability, we followed the approach in ref. 24. Briefly, we incubated the HA pseudovirus library in different acidic pH buffers before infecting MDCK-SIAT1 cells. We also included a condition where the HA pseudovirus library was incubated with neutral pH media before infection.

A total of $7 \times 10^5$ MDCK-SIAT1 cells in NAM were plated in each well of six-well plates. The 2.5 µg ml$^{-1}$ of amphotericin B was added to the MDCK-SIAT1 cells when plating. On the next day, aliquots containing ~$2.4 \times 10^6$ TU ml$^{-1}$ of HA pseudovirus library were incubated with citrate-based acidic buffers at pH 6.1, 5.9, 5.7, 5.5, 5.3 or NAM (neutral pH condition) for 60 min at 37 °C. After incubation, libraries were concentrated with 100,000 Amicon spin columns (Millipore, UFC910008) by spinning for 15 min at 1,500g, resuspending in 11 ml of PBS to neutralize the acidic buffers and spun down again for 20 min at 1,500g. The libraries were then resuspended in 2 ml of NAM, treated with 500 nM of oseltamivir for 20 min on ice to inhibit NA and used to infect the plated MDCK-SIAT1 cells.

At 12 h after infection, the non-integrated reverse-transcribed lentiviral genomes were recovered by miniprepping the MDCK-SIAT1 cells. In the miniprep lysis step where P2 buffer is added, we spiked in a DNA standard at an amount calculated to be approximately 3% of the recovered lentiviral DNA (based on the estimated number of non-integrated lentiviral genomes) under normal infection conditions with no acidic buffer treatment. The rationale for including the DNA spike-in standard is to enable relative sequencing counts to be converted into absolute quantities of each barcoded pseudovirus variant, normalized to the standard, across different acidic buffer conditions. This DNA standard is a plasmid that encodes the lentiviral backbone with a barcoded mCherry gene; the plasmid map is 3068_ForInd_mC_BCs_pool1 and the barcodes are available via GitHub at https://github.com/dms-vep/Flu_H3_Massachusetts2022_DMS/blob/main/data/neutralization_standard_barcodes.csv. Afterwards, amplicons for Illumina sequencing were prepared as described in the previous section. For details on how sequencing counts were converted to mutation effects on acid stability, see the section below on 'Illumina sequencing barcode analysis'.

## Mutation effects on sera neutralization

To measure effects of HA mutations on sera neutralization, we followed the approach in ref. 23. A total of $7 \times 10^5$ MDCK-SIAT1 cells in NAM were plated in each well of six-well plates. A total of 2.5 µg ml$^{-1}$ of amphotericin B was added to the MDCK-SIAT1 cells when plating. On the next day, aliquots containing ~$1.2 \times 10^6$ TU ml$^{-1}$ of HA pseudovirus library were treated with 500 nM oseltamivir and incubated with three concentrations of sera estimated to span between IC98 and IC98*16. These inhibitory concentration values were determined by a luciferase-based pseudovirus neutralization assay. Several dilutions of sera are necessary for improving estimation of mutation effects on sera neutralization. Libraries and sera were incubated for 60 min at 37°C. After incubation, libraries were used to infect the plated MDCK-SIAT1 cells. At 12 h after infection, the non-integrated reverse-transcribed lentiviral genomes were recovered by miniprepping with the spike-in DNA standard and amplicons for Illumina sequencing were prepared as described in the previous section. For details on how sequencing counts were converted to mutation effects on sera neutralization, see the section below on 'Illumina sequencing barcode analysis'.

## Production of conditionally replicative influenza viruses

Conditionally replicative influenza viruses that lack the PB1 gene were produced by reverse genetics[29,70]. The native HA sequence was cloned into a bidirectional pHW2000 influenza reverse genetics plasmid. The plasmid map for A/Massachusetts/18/2022 HA is 5012_pHW_MA22_HA and the plasmid map for A/Perth/16/2009 HA is 1442_pHWPerth09_HA. Mutant HA plasmids were cloned by PCR with partially overlapping primers that contain the mutation of interest, followed by HiFi assembly. All plasmids were sequence confirmed by Plasmidsaurus. To perform the virus rescue, $5 \times 10^5$ 293T-CMV-PB1 cells and $4 \times 10^5$ MDCK-SIAT1-CMV-PB1-TMPRSS2 cells were plated in D10 in each well of six-well plates. On the next day, each well was transfected with 2 µg of total of plasmids including: 0.25 µg each of six reverse genetics plasmids expressing genes from A/WSN/1933 (30_pHW181_PB2, 32_pHW183_PA, 34_pHW185_NP, 35_pHW186_NA, 36_pHW187_M and

37_pHW188_NS), 0.25 μg of plasmid that expresses eGFP in place of PB1 (208_pHH_PB1flank_eGFP) and 0.25 μg of HA reverse genetics plasmid. At 24 h after transfection, D10 was aspirated and each well was replenished with 2 ml of IGM. At 48 h after media swap, viral supernatant was spun down for 4 min at 845g and aliquots of clarified supernatant were collected and frozen down at −80 °C.

### Validation of cell entry effects
Conditionally replicative influenza viruses were serially diluted in NAM in 96-well plates. A total of $5 \times 10^4$ MDCK-SIAT1-CMV-PB1 cells in NAM were added to each well. Note that these cells do not express TMPRSS2, so the influenza viruses can only undergo a single cycle of infection. At 16 h after infection, wells with 1–10% eGFP-positive cells were selected. Precise measurements of the percentage of eGFP-positive cells in these wells were obtained by flow cytometry and viral titre were calculated using a Poisson distribution.

### Validation of acid stability effects
A total of $1.5 \times 10^5$ MDCK-SIAT1-CMV-PB1 cells in D10 were plated in each well of a 12-well plate. Conditionally replicative influenza viruses were diluted to a target MOI of 0.5–1 (~2–10 μl of virus) in 100 μl of citrate-based acidic buffers at pH 5.7, 5.5 and 5.3 or NAM (neutral pH condition) and incubated for 60 min at 37 °C. The pH-treated viruses were then brought back to neutral pH by diluting the 100 μl into 2 ml of NAM. At 4 h after plating the MDCK-SIAT1-CMV-PB1 cells, D10 was aspirated and cells were washed with 1 ml of 1× PBS before 2 ml of the NAM-diluted viruses were added. At 16 h after infection, the percentage of eGFP-positive cells in each well was determined by flow cytometry. The fraction infectivity retained was calculated as the ratio of percentage eGFP-positive cells when virus was treated with acidic pH over the percentage of eGFP-positive cells when virus was treated with NAM.

### Neutralization assays
Sera were serially diluted in NAM in 96-well plates. Conditionally replicative influenza viruses were diluted to a target MOI that falls within a range where the fluorescence signal would change linearly with respect to neutralization. The virus and sera dilutions were incubated for 60 min at 37 °C. Afterwards, $4 \times 10^4$ MDCK-SIAT1-CMV-PB1 cells were added to each well. At 16 h after infection, the fluorescence signal was read on a Tecan M1000 plate reader and the fraction infectivity was determined relative to no serum controls.

### PacBio sequencing analysis
PacBio circular consensus sequences (CCSs) were aligned to the HA reference sequence using alignparse[71]. CCSs for each barcode were determined by requiring at least three CCSs per barcode. The final barcode-variant table is available via GitHub at https://github.com/dms-vep/Flu_H3_Massachusetts2022_DMS/blob/main/results/variants/codon_variants.csv.

For full details on the analysis, see the following notebooks for:

- Analysing the PacBio CCSs: https://dms-vep.org/Flu_H3_Massachusetts2022_DMS/notebooks/analyze_pacbio_ccs.html
- Building PacBio consensus sequences: https://dms-vep.org/Flu_H3_Massachusetts2022_DMS/notebooks/build_pacbio_consensus.html
- Building the final barcode-variant table: https://dms-vep.org/Flu_H3_Massachusetts2022_DMS/notebooks/build_codon_variants.html

### Illumina sequencing barcode analysis
From the Illumina short-read sequencing data, barcodes were counted by https://jbloomlab.github.io/dms_variants/dms_variants.illumina-barcodeparser.html and then mutation effects were calculated using approaches described previously[23,24] and outlined below.

To convert barcode counts into mutation effects on cell entry, we first calculated functional scores. Briefly, a functional score for a variant $v$ was calculated as $\log_2[(n_{post}^v/n_{post}^{wt})/(n_{pre}^v/n_{pre}^{wt})]$, where each $n$ is a count of barcodes that entered cells. Specifically, $n_{post}^v$ is the count of each variant in the HA pseudovirus library, $n_{pre}^v$ is the count of each variant in the VSV-G pseudovirus library, $n_{post}^{wt}$ and $n_{pre}^{wt}$ are counts of the unmutated (wildtype) variants in these libraries. Positive functional scores indicate that the variant is better at entering cells relative to the unmutated HA, while negative functional scores indicate that the variant is worse at entering cells relative to the unmutated HA. Since some variants contain several mutations, we used multidms[72] (https://matsengrp.github.io/multidms) to fit a global epistasis model with a sigmoid function using the functional scores to obtain individual mutation effects on cell entry. For more details on fitting, see the notebooks under 'Functional effects of mutations' at https://dms-vep.org/Flu_H3_Massachusetts2022_DMS/appendix.html. We report the median mutation effect across library replicates and filter for mutations that are seen in at least two different barcoded variants (averaged across libraries). See https://dms-vep.org/Flu_H3_Massachusetts2022_DMS/cell_entry.html for interactive visualizations of mutation effects on cell entry.

To convert barcode counts into mutation effects on acid stability and sera neutralization, we calculated the fraction infectivity of each variant retained at each acidic pH buffer treatment or serum concentration, normalizing to the counts of spike-in standard barcodes in each condition. We then fit a biophysical model to these fractional infectivity data using polyclonal[27] (https://jbloomlab.github.io/polyclonal) to obtain individual mutation effects on acid stability and sera neutralization. For more details on fitting, see the notebooks under 'Antibody/serum escape' and 'Stability' at https://dms-vep.org/Flu_H3_Massachusetts2022_DMS/appendix.html. We report the average mutation effect across library replicates and filter for mutations that are seen in at least two different barcoded variants (averaged across libraries) and have a cell entry score >−3. See https://dms-vep.org/Flu_H3_Massachusetts2022_DMS/acid_stability.html for interactive visualizations of mutation effects on acid stability and https://dms-vep.org/Flu_H3_Massachusetts2022_DMS/sera_neutralization.html for interactive visualizations of mutation effects on sera neutralization.

### Entropy calculation from natural sequences
The subsampled Nextstrain tree was obtained from ref. 73. The subsampling approach accounts for biases through evenly sampling sequences by year and major geographical region. This H3N2/HA/60y build is available at https://nextstrain.org/groups/blab/flu/seasonal/h3n2/ha/60y.

We calculate entropy from the amino acid frequencies at a given position. These frequencies are derived from the number of tips in the Nextstrain tree with a given amino acid, divided by the total number of tips in the Nextstrain tree. For example, consider a site where only two amino acids have been observed, with $X$ tips of amino acid A and $Y$ tips of amino acid B. The total number of tips in the tree is $N = X + Y$. Then, the entropy can be calculated using scipy.stats as: entropy([$X/N$, $Y/N$]).

### Evolutionary entrenchment analysis
The frequencies of amino acids at different timepoints were obtained from the H3N2/HA/60y Nextstrain tree. An amino acid was considered fixed if at any timepoint its frequency at a given site was >95%. Sites were considered inside the receptor-binding pocket if they were within 4 Å of sialic acid or previously reported to affect receptor binding[74–76]. See Extended Data Table 1 for the full definition of receptor-binding pocket sites. See https://dms-vep.org/Flu_H3_Massachusetts2022_DMS/entrenchment.html for an interactive plot of the analysis.

### Structural analysis
UCSF ChimeraX v.1.8 (ref. 77) was used for structural visualizations. All Protein Data Bank accession IDs used are included in Figs. 1 and 2 and Extended Data Fig. 8.

## Reporting summary

Further information on research design is available in the Nature Portfolio Reporting Summary linked to this article.

## Data availability

Data that have been prefiltered for quality control criteria are available in CSV format at these links: mutation effects on cell entry and acid stability via GitHub at https://github.com/dms-vep/Flu_H3_Massachusetts2022_DMS/blob/main/results/summaries/Phenotypes.csv and mutation effects on sera neutralization via GitHub at https://github.com/dms-vep/Flu_H3_Massachusetts2022_DMS/blob/main/results/summaries/Phenotypes_per_antibody_escape.csv. Raw sequencing data are available under BioProject PRJNA1320726 in the NCBI Sequence Read Archive. Source data are provided with this paper.

## Code availability

See https://dms-vep.org/Flu_H3_Massachusetts2022_DMS for a collection of interactive visualizations. Code for reproducing the analysis is available via GitHub at https://github.com/dms-vep/Flu_H3_Massachusetts2022_DMS and output of the analysis is at https://dms-vep.org/Flu_H3_Massachusetts2022_DMS/appendix.html.

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

## Acknowledgements

This work was supported in part by the NIH/NIGMS CMB training grant (T32 GM007270) to T.C.Y., NSF Graduate Research Fellowship (DGE-2140004) to T.C.Y., NIH/NIAID under award R01AI165821 to J.D.B. and NIH/NIAID under contract 75N93021C00015 to J.D.B. J.D.B. is an investigator of the Howard Hughes Medical Institute. This research was also supported by the Genomics & Bioinformatics Shared Resource (RRID: SCR_022606), the Flow Cytometry Shared Resource (RRID: SCR_022613) of the Fred Hutch/University of Washington/Seattle Children's Cancer Consortium (P30 CA015704) and by Fred Hutch Scientific Computing, NIH grants S10-OD-020069 and S10-OD-028685. We thank the Bedford laboratory at Fred Hutch for maintaining Nextstrain builds, J. Huddleston for technical assistance and B. Larsen for help with data visualization. This paper is the result of funding in whole or in part by the National Institutes of Health (NIH). It is subject to the NIH Public Access Policy. Through acceptance of this federal funding, NIH has been given a right to make this paper publicly available in PubMed Central upon the Official Date of Publication, as defined by NIH.

## Author contributions

T.C.Y. and J.D.B. conceived the study. T.C.Y. and C.K. performed the experiments. T.C.Y. and J.D.B. performed the computational analysis. T.C.Y., C.K., B.D., A.N.L. and J.D.B. interpreted the results. T.C.Y. and J.D.B. wrote the original draft. J.A.E. provided resources. All authors edited and approved the paper.

## Competing interests

J.D.B. consults on topics related to viral evolution for Apriori Bio, Invivyd, the Vaccine Company, Pfizer and GSK. J.D.B., B.D. and A.N.L. are inventors on Fred Hutch licensed patents related to viral deep mutational scanning. The other authors declare no competing interests.

## Additional information

**Extended data** is available for this paper at https://doi.org/10.1038/s41559-025-02895-1.

**Correspondence and requests for materials** should be addressed to Jesse D. Bloom.

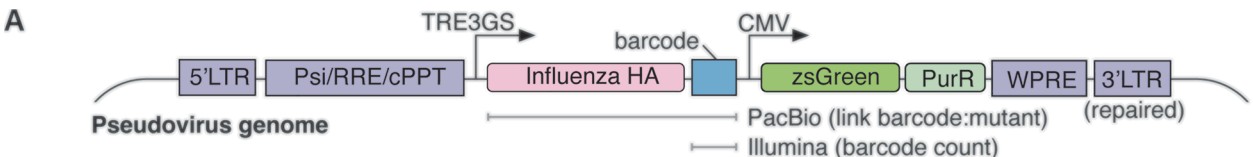

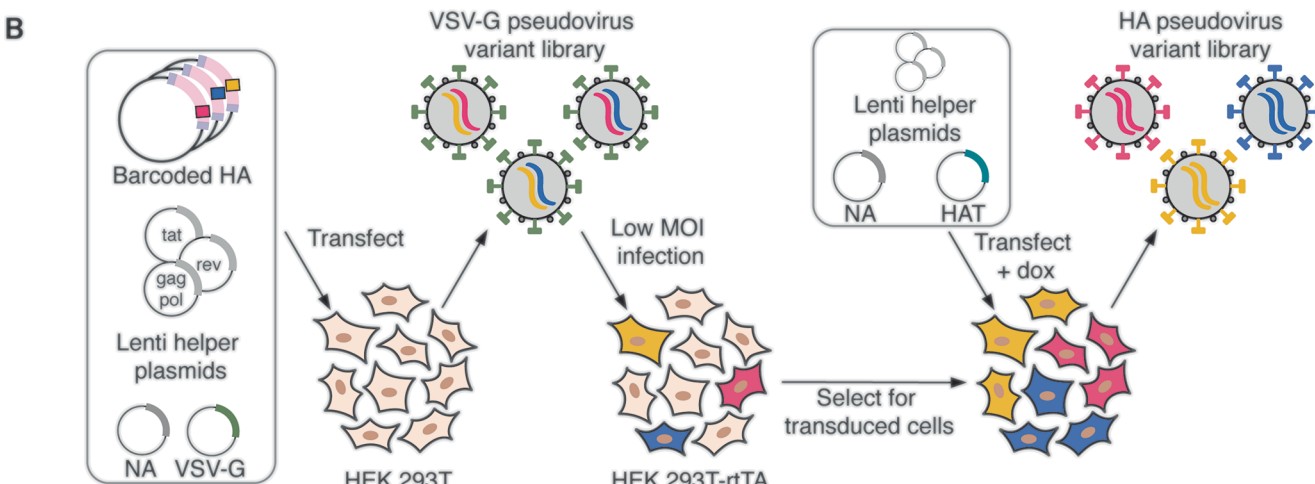

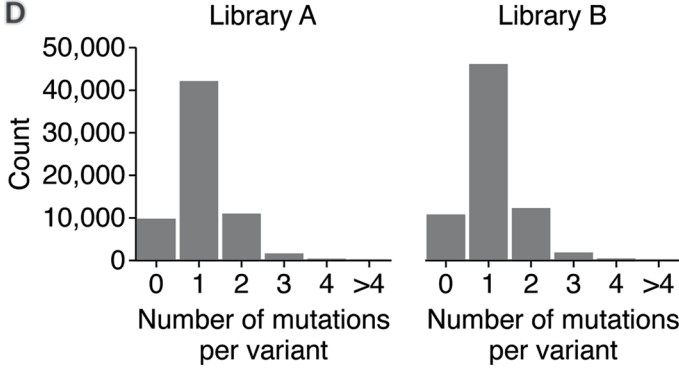

**Extended Data Fig. 1 | Pseudovirus deep mutational scanning of influenza hemagglutinin. A**) Diagram of the lentiviral genome used to produce genotype-phenotype-linked pseudovirus libraries for deep mutational scanning. The genome is flanked by long terminal repeat (LTR) sequences, with the typical 3' LTR deletion repaired so the lentiviral genome can be transcribed after integration. A zsGreen reporter and a puromycin resistance marker are constitutively expressed from a CMV promoter. Expression of the HA gene is regulated by a doxycycline inducible TRE3GS promoter. PacBio sequencing is performed to map each barcode to an HA mutant. Then, effects of HA mutations can be quantified by Illumina sequencing the barcodes. **B**) Schematic of the "two-step" method for generating genotype-phenotype-linked pseudovirus libraries described in Dadonaite et al.[23,24]. In the first step, a plasmid library encoding the lentiviral genomes with the HA mutants is co-transfected into HEK293T cells alongside three lentiviral helper plasmids (tat, rev, gagpol), a plasmid expressing a strain-matched neuraminidase (NA), and a plasmid expressing the glycoprotein from vesicular stomatitis virus (VSV-G). This results in pseudoviruses that encode HA mutants within their genomes but express VSV-G and NA on their surfaces. The NA ensures HA expression does not prevent virions from detaching from producing cells. These VSV-G pseudotyped viruses are transduced into a HEK293T-rtTA cell line at low MOI to ensure most infected cells integrate a single lentiviral genome, and puromycin is used to select for integrated cells. In the second step, helper plasmids, a plasmid expressing NA, and a plasmid expressing the HA-activating human airway trypsin-like (HAT) protease are co-transfected into the integrated cells. Doxycycline is added at this step to induce HA expression. This results in genotype-phenotype-linked HA-pseudotyped pseudoviruses. These pseudoviruses can undergo a single round of cell entry, but are not fully infectious agents as they do not encode the genes needed to undergo multiple rounds of replication. **C**) Number of barcodes and mutation coverage in the two pseudovirus library replicates. In this table, "% mutations present" indicates the percentage of all HA ectodomain amino-acid mutations found in at least one of the barcoded variants. **D**) Distribution of the number of HA amino-acid mutations per variant in the two pseudovirus library replicates. Most variants contain a single mutation.

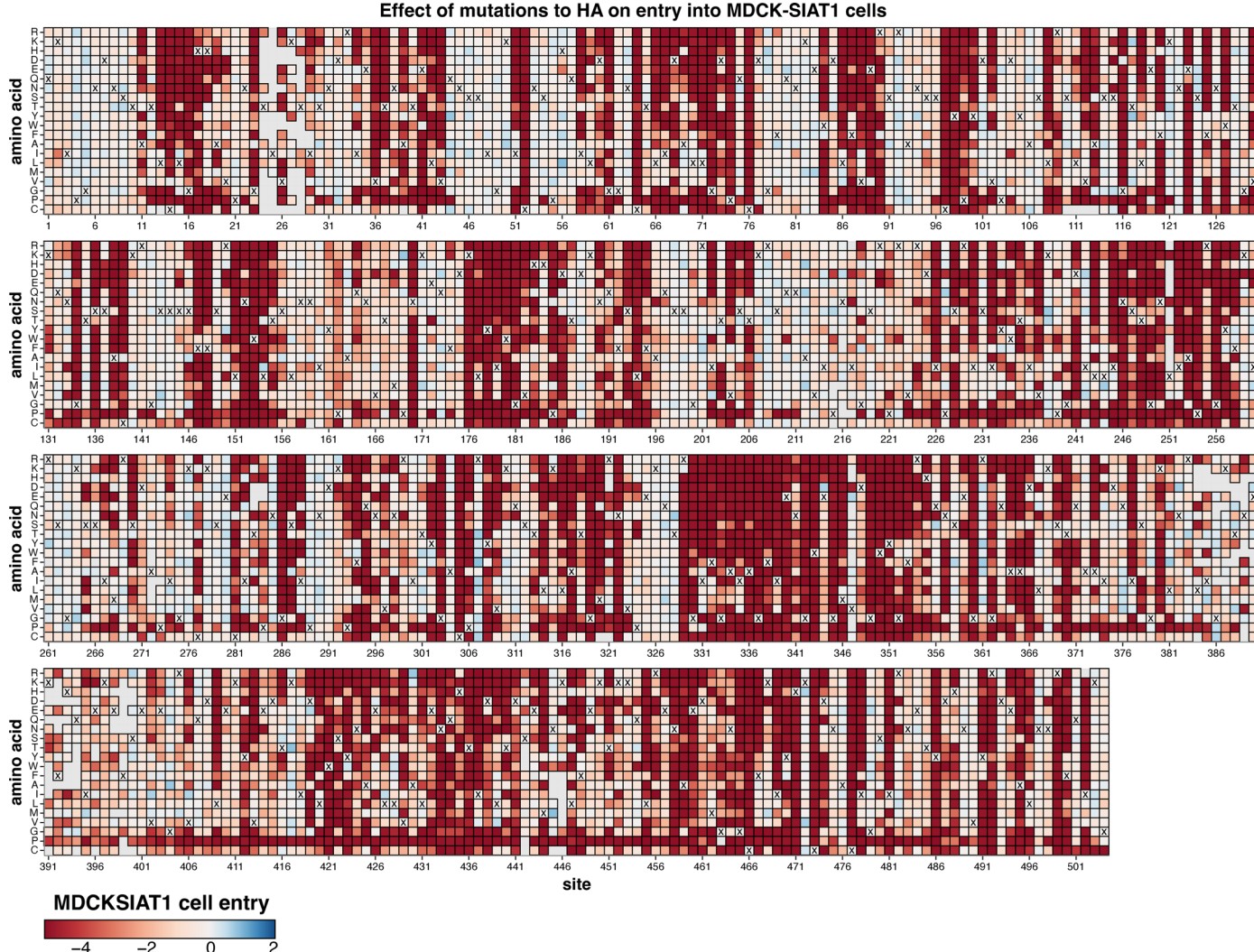

**Effect of mutations to HA on entry into MDCK-SIAT1 cells**

**MDCKSIAT1 cell entry**

−4    −2    0    2

**Extended Data Fig. 2 | Mutation effects on HA-mediated cell entry.** Each tile represents a mutation at an HA site, colored by the effect of that mutation on entry into MDCK-SIAT1 cells. Red indicates impaired entry, white indicates no effect, and blue indicates improved entry. To visualize these mutation effects in the context of the HA structure, see Fig. 1b. Tiles with an 'X' denote the amino acid identity in the unmutated MA22 strain. Empty gray tiles indicate mutations that were either missing from the library or lacked a reliable cell entry measurement. See https://dms-vep.org/Flu_H3_Massachusetts2022_DMS/cell_entry.html for an interactive version of this heatmap.

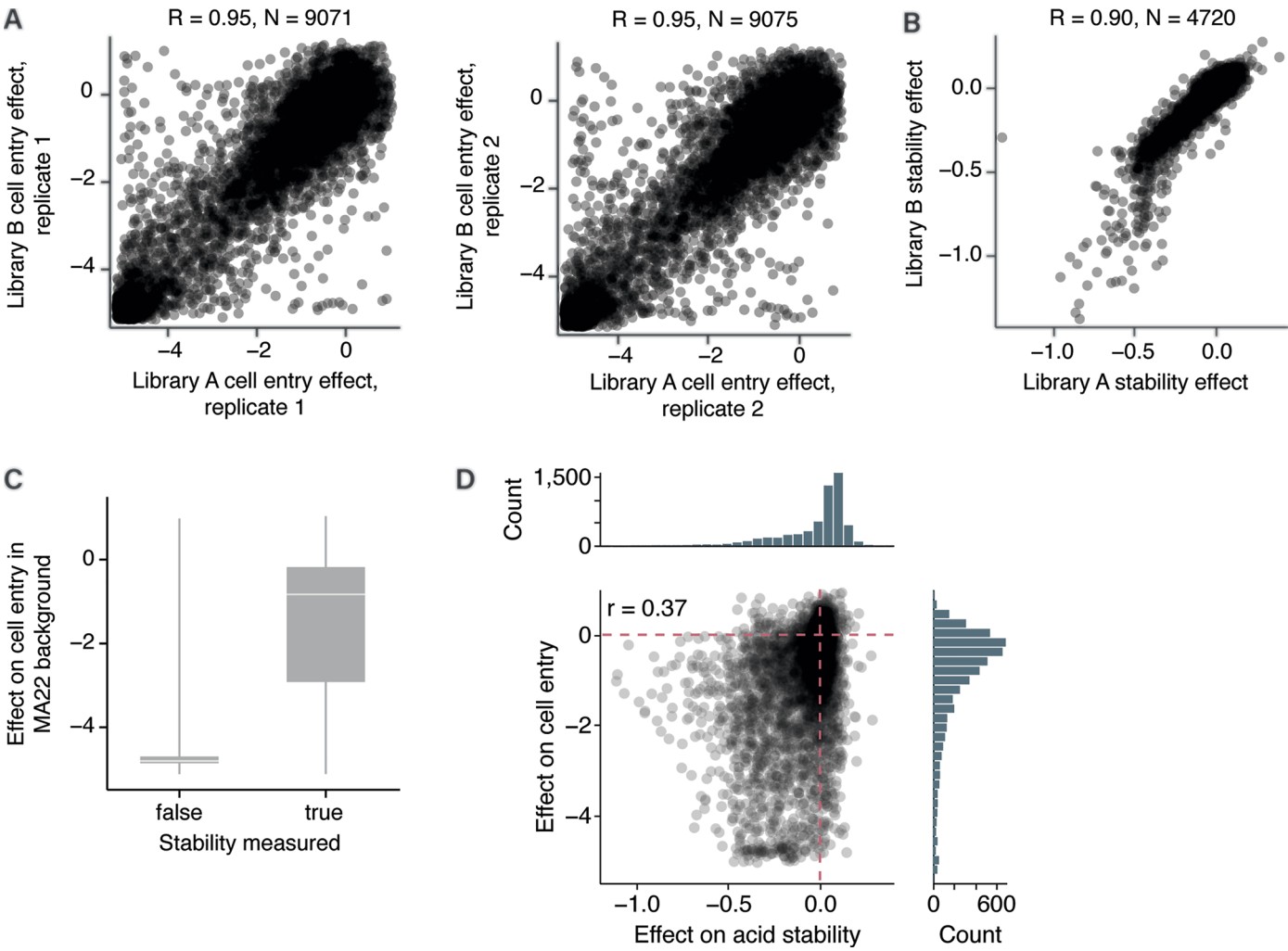

**Extended Data Fig. 3 | Correlations of mutation effects measured by deep mutational scanning. A**) Correlation of the effects of HA mutations on cell entry between the two pseudovirus library replicates. Each point represents the effect of a different mutation as measured in each replicate. Two technical replicates were performed for each of the two libraries, and the two panels show correlations between the two independent libraries for each technical replicate. Throughout this paper, we report the median effect of mutations across the four replicates. **B**) Correlation of the effects of HA mutations on acid stability between the two pseudovirus library replicates. Note that there are fewer mutations with measured effects on stability because we can only measure stability for mutations with at least some cell entry. **C**) Most mutations for which it was not possible to make measurements of acid stability correspondingly have very poor cell entry. The center line shows the median effect on cell entry, the box indicates the interquartile range, and the whiskers extend 1.5 x interquartile range beyond the first and third quartiles. **D**) Correlation between effects of mutations on cell entry and effects of mutations on acid stability. These effects of mutations on these two phenotypes are only weakly correlated.

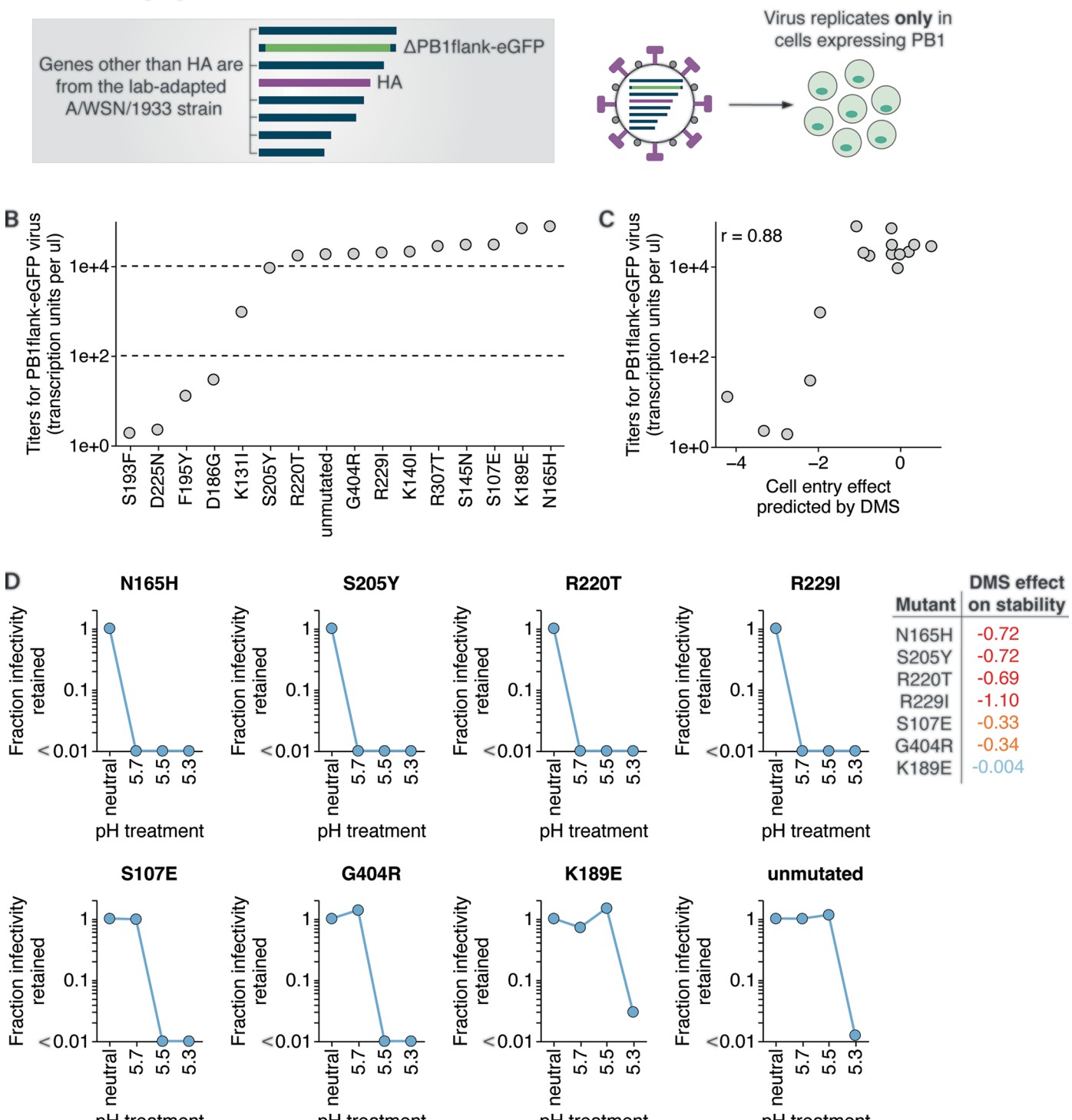

**Extended Data Fig. 4 | Validation of mutation effects on cell entry and acid stability with conditionally replicative influenza virions. A**) Diagram of a conditionally replicative PB1flank-eGFP influenza virus genome. The PB1 gene is replaced with an eGFP and the remaining segments besides HA are derived from a lab-adapted A/WSN/1933 strain. These viruses can be rescued by reverse genetics, can only replicate in cells that express PB1, and are safe to use at biosafety-level 2. **B**) Titers of conditionally replicative virions carrying single amino acid mutations in the MA22 HA that were present in the supernatant after virus production by reverse genetics. Each point in the plot is the mean of four titer measurements, two technical replicates from the same virion rescue stock and two biological replicate stocks rescued from independent plasmid preparations. **C**) Correlation between the titers of conditionally replicative virions carrying single amino acid mutations to the MA22 HA (shown in B) and the effects of those mutations on cell entry measured by deep mutational scanning. **D**) The fraction infectivity retained after treating conditionally replicative virions with the indicated MA22 HA mutations with either neutral media or acidic pH buffers. The fractions are normalized to the infectivity in the neutral condition. Each point is the mean of two technical replicates performed on different days. The effect of each mutant on acid stability measured by deep mutational scanning is included on the right, and generally tracks with the pH sensitivity measured in the validation assay.

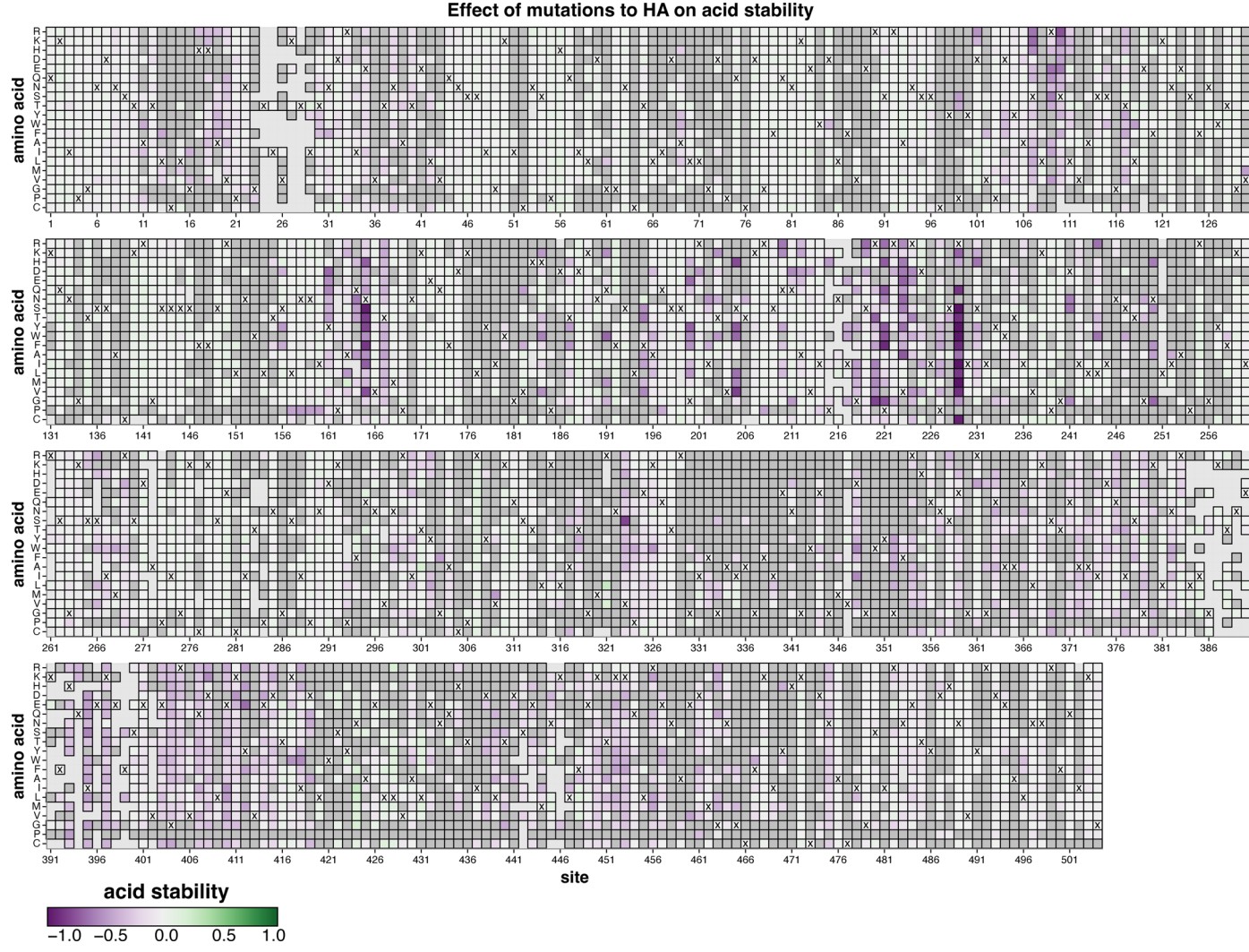

**Extended Data Fig. 5 | Mutation effects on HA acid stability.** Each tile represents a mutation at an HA site, colored by the effect of that mutation on HA acid stability. Purple indicates decreased stability, white indicates no effect, and green indicates increased stability. To visualize these mutation effects in the context of the HA structure, see Fig. 2c. Tiles with an 'X' denote the amino acid identity in the unmutated MA22 strain. Dark gray tiles indicate mutations that are too deleterious for cell entry to reliably measure their effect on acid stability, while light gray tiles indicate mutations that were missing (not measured) in the library. See https://dms-vep.org/Flu_H3_Massachusetts2022_DMS/acid_stability.html for an interactive version of this heatmap.

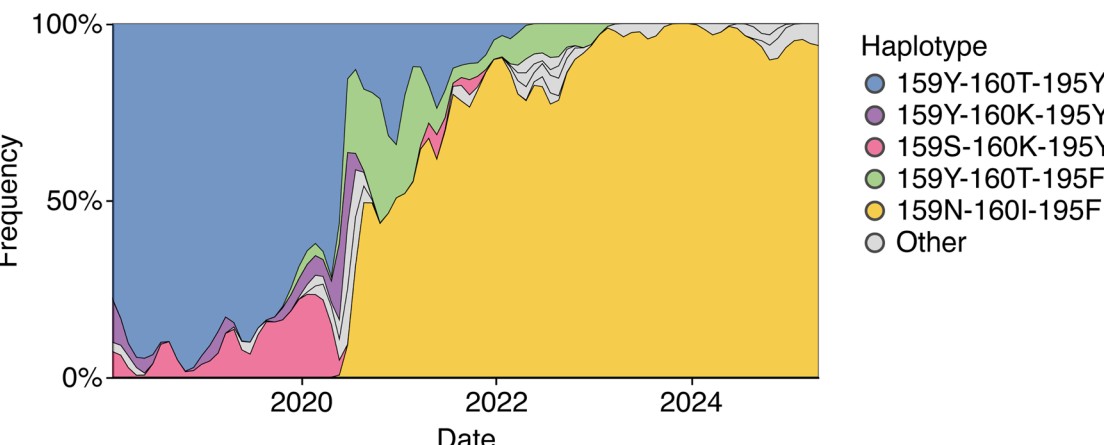

**Extended Data Fig. 6 | Evolutionary dynamics at sites 159, 160, and 195 in human H3N2 HA.** Muller diagram showing all combinations of amino acids observed at sites 159, 160, and 195 since 2018 in the evolution of the HA from human H3N2 influenza. Haplotypes that reach a frequency >20% at some timepoint are colored according to the key, while other haplotypes are colored gray. Y159N and T160I only arise to fixation in the background of 195F, and this triple mutant lineage (yellow) eventually outcompetes the lineage containing 195F alone (light green).

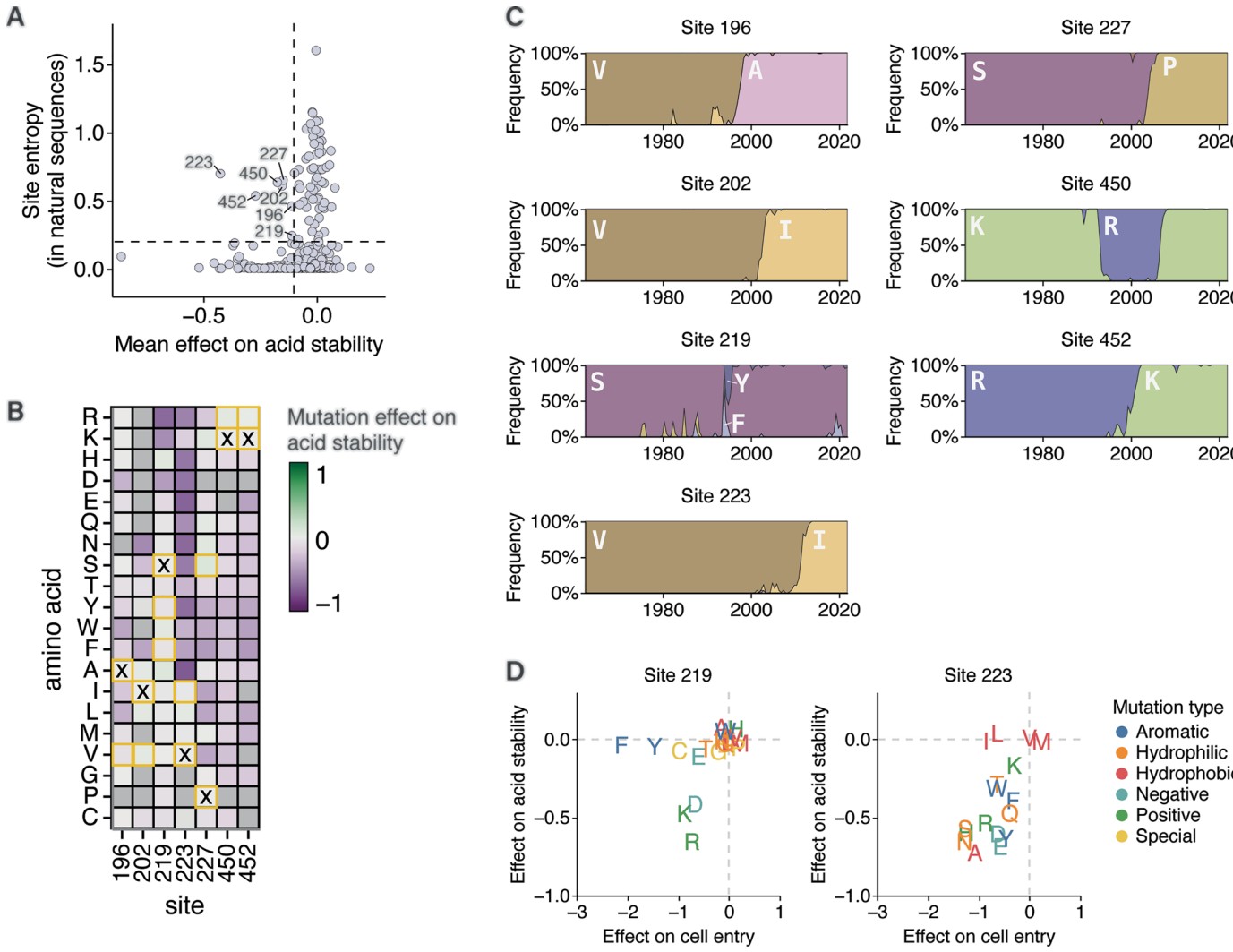

**Extended Data Fig. 7 | Conservation at HA sites with destabilizing mutations.** **A**) Correlation between the Shannon entropy at each HA site in an alignment from a subsampled tree of natural human H3N2 evolution since 1968 and the mean effect on acid stability of all mutations at each HA site as measured by deep mutational scanning. Most sites with many mutations that destabilize HA are strongly conserved, with exceptions labeled. **B**) Heatmap of all mutation effects on acid stability at sites labeled in A. The 'X' indicates the amino acid in the unmutated MA22 HA, and squares boxed in yellow indicate amino acids that increased in frequency during natural evolution. At sites with many destabilizing mutations, natural evolution exclusively samples only the (relatively rare) amino-acid identities that do not destabilize HA. **C**) Frequencies over time (the x-axis indicates year) of amino acids observed in natural human H3N2 sequences at the sites labeled in A. **D**) Correlation between mutation effects on acid stability and cell entry at sites 219 and 223. Mutations are colored by biochemical group.

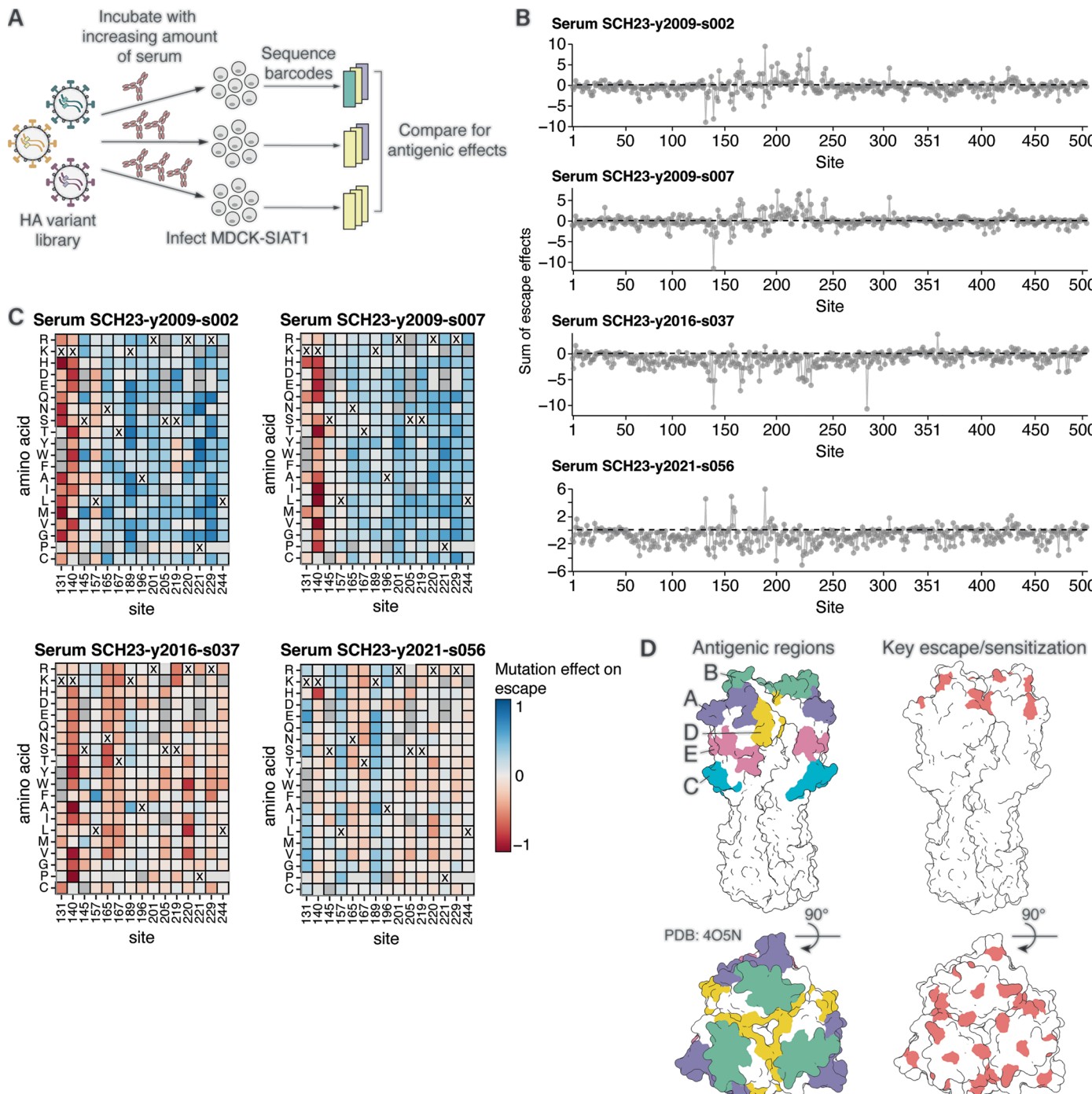

**Extended Data Fig. 8 | Mapping the effects of HA mutations on neutralization by human sera using deep mutational scanning. A)** We incubated the pseudovirus HA library with increasing concentrations of serum, then infected MDCK-SIAT1 cells and sequenced the barcodes of pseudoviruses that were still able to enter cells after serum treatment. To quantify mutation effects on antigenicity, we compared these barcode counts to those of pseudoviruses that were not incubated with serum, which served as an infection baseline; a neutralization standard is used to convert these counts into fraction infectivity at each serum concentration (**Methods**)[23,24]. The mutation effects we report are the median of two biological replicates. **B)** The sum of mutation effects on escape at each site in HA for four human sera. Fig. 4a shows these four plots overlaid. See https://dms-vep.org/Flu_H3_Massachusetts2022_DMS/sera_neutralization.html for a version of these lineplots that is interactive, along with interactive heatmaps that show how individual mutations affect sera neutralization. **C)** Mutation-level escape and sensitization at key sites that are highlighted in Fig. 4a for the four sera. The X's indicate the amino acid in the unmutated MA22 strain. Tiles that are more blue indicate mutations that escape sera, while tiles that are more red indicate mutations that have a sensitizing effect. **D)** HA structures (Protein Data Bank 4O5N) showing antigenic regions and locations of key sites of escape or sensitization that are highlighted in Fig. 4a.

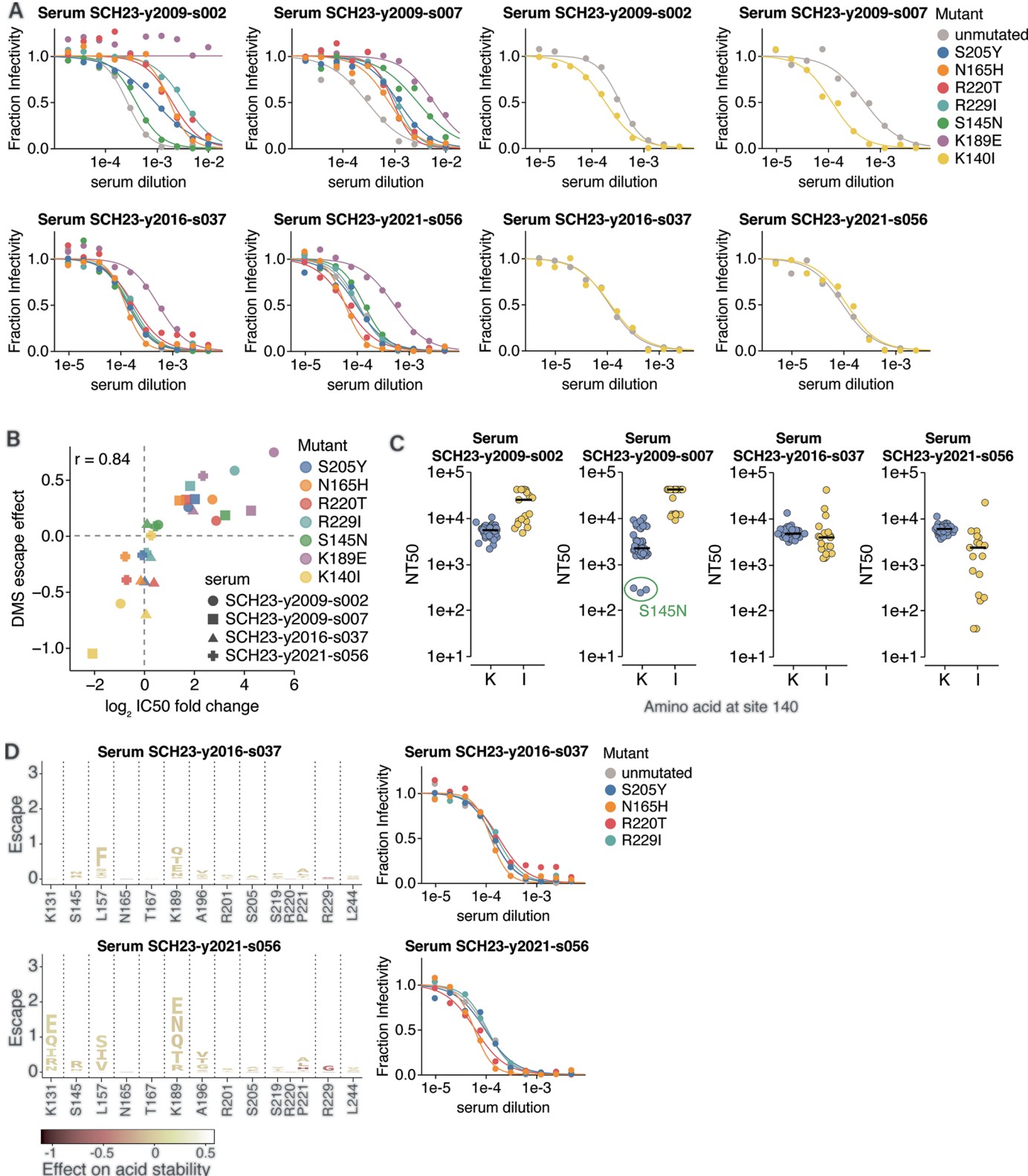

**Extended Data Fig. 9 | See next page for caption.**

**Extended Data Fig. 9 | Validation of mutation effects on serum neutralization with conditionally replicative influenza virions. A**) Neutralization of conditionally replicative influenza virions with the MA22 HA carrying the indicated mutations (S205Y, N165H, R220T, R229I, S145N, K189E, and K140I) by four human sera collected in 2023. Each point is the mean of two technical replicates. A subset of these curves are shown in Fig. 4c, Fig. 4f, and Extended Data Fig. 9D. **B**) Correlation between the antigenic effect of mutations measured by deep mutational scanning and the change in IC50 measured by the independent neutralization assay in A. **C**) Neutralization for 78 vaccine or circulating H3N2 strains between 2012 and 2023 by the four sera, as measured in Kikawa et al.[3]. The points are the median of two or three barcoded replicates and are stratified by whether or not the strain includes a K or I at site 140.

The black line indicates the median NT50. Two of the sera have higher titers against strains with 140I, consistent with K140I being a sensitizing mutation for these sera. There are three recent strains that contain S145N that escape SCH23-y2009-s007, consistent with this being an escape mutation from this sera. **D**) Logoplots displaying single nucleotide accessible mutations from MA22 HA with positive escape at the key sites highlighted red in Fig. 4a for two sera. The height of each letter is proportional to the escape from the indicated serum as measured by deep mutational scanning. Each mutation is colored by its effect on HA acid stability as measured in the deep mutational scanning, with darker colors indicating decreased stability. The logoplots and neutralization curves for the other two sera mapped by deep mutational scanning are shown in Fig. 4e and Fig. 4f. In the neutralization curves, each point is the mean of two technical replicates.

**Extended Data Table 1 | Sites within receptor binding pocket and antigenic regions**

| Receptor binding pocket region | Sites |
|---|---|
| 130-loop | 128, 130, 131, 133-138 |
| 150-loop | 155-160 |
| 190-helix | 186, 189-194, 196-198 |
| 220-loop | 221-228 |
| Base | 98, 153, 183, 195 |
| Other | 145 |
| **Antigenic region** | |
| Epitope A | 122-146 |
| Epitope B | 155-160, 186-198 |
| Epitope C | 44-54, 273-280 |
| Epitope D | 166-181, 201-219 |
| Epitope E | 62-65, 78-94, 260-265 |

# Reporting Summary

## Statistics

For all statistical analyses, confirm that the following items are present in the figure legend, table legend, main text, or Methods section.

| n/a | Confirmed | |
|---|---|---|
| ☐ | ☒ | The exact sample size (*n*) for each experimental group/condition, given as a discrete number and unit of measurement |
| ☐ | ☒ | A statement on whether measurements were taken from distinct samples or whether the same sample was measured repeatedly |
| ☒ | ☐ | The statistical test(s) used AND whether they are one- or two-sided<br>*Only common tests should be described solely by name; describe more complex techniques in the Methods section.* |
| ☒ | ☐ | A description of all covariates tested |
| ☒ | ☐ | A description of any assumptions or corrections, such as tests of normality and adjustment for multiple comparisons |
| ☐ | ☒ | A full description of the statistical parameters including central tendency (e.g. means) or other basic estimates (e.g. regression coefficient) AND variation (e.g. standard deviation) or associated estimates of uncertainty (e.g. confidence intervals) |
| ☒ | ☐ | For null hypothesis testing, the test statistic (e.g. *F*, *t*, *r*) with confidence intervals, effect sizes, degrees of freedom and *P* value noted<br>*Give P values as exact values whenever suitable.* |
| ☒ | ☐ | For Bayesian analysis, information on the choice of priors and Markov chain Monte Carlo settings |
| ☒ | ☐ | For hierarchical and complex designs, identification of the appropriate level for tests and full reporting of outcomes |
| ☒ | ☐ | Estimates of effect sizes (e.g. Cohen's *d*, Pearson's *r*), indicating how they were calculated |

*Our web collection on statistics for biologists contains articles on many of the points above.*

## Software and code

Policy information about availability of computer code

| | |
|---|---|
| Data collection | No specific software was used in data collection. |
| Data analysis | Data analysis code is available at:<br>https://github.com/dms-vep/Flu_H3_Massachusetts2022_DMS<br>Most of the analysis in the Github repository is performed using dms-vep-pipeline-3 (https://github.com/dms-vep/dms-vep-pipeline-3, version 3.14.1).<br>Flow cytometry data analysis was performed using FlowJo (version 10.7.2).<br>Structural analysis was performed using ChimeraX (version 1.8). |

For manuscripts utilizing custom algorithms or software that are central to the research but not yet described in published literature, software must be made available to editors and reviewers. We strongly encourage code deposition in a community repository (e.g. GitHub). See the Nature Portfolio guidelines for submitting code & software for further information.

## Data

Policy information about availability of data

All manuscripts must include a data availability statement. This statement should provide the following information, where applicable:

- Accession codes, unique identifiers, or web links for publicly available datasets
- A description of any restrictions on data availability
- For clinical datasets or third party data, please ensure that the statement adheres to our policy

> Raw sequencing data has been uploaded under BioProject PRJNA1320726.
>
> Data generated by the analysis is available at: https://github.com/dms-vep/Flu_H3_Massachusetts2022_DMS/tree/main/results

## Research involving human participants, their data, or biological material

Policy information about studies with human participants or human data. See also policy information about sex, gender (identity/presentation), and sexual orientation and race, ethnicity and racism.

| | |
|---|---|
| Reporting on sex and gender | *Use the terms sex (biological attribute) and gender (shaped by social and cultural circumstances) carefully in order to avoid confusing both terms. Indicate if findings apply to only one sex or gender; describe whether sex and gender were considered in study design; whether sex and/or gender was determined based on self-reporting or assigned and methods used. Provide in the source data disaggregated sex and gender data, where this information has been collected, and if consent has been obtained for sharing of individual-level data; provide overall numbers in this Reporting Summary. Please state if this information has not been collected. Report sex- and gender-based analyses where performed, justify reasons for lack of sex- and gender-based analysis.* |
| Reporting on race, ethnicity, or other socially relevant groupings | *Please specify the socially constructed or socially relevant categorization variable(s) used in your manuscript and explain why they were used. Please note that such variables should not be used as proxies for other socially constructed/relevant variables (for example, race or ethnicity should not be used as a proxy for socioeconomic status). Provide clear definitions of the relevant terms used, how they were provided (by the participants/respondents, the researchers, or third parties), and the method(s) used to classify people into the different categories (e.g. self-report, census or administrative data, social media data, etc.) Please provide details about how you controlled for confounding variables in your analyses.* |
| Population characteristics | *Describe the covariate-relevant population characteristics of the human research participants (e.g. age, genotypic information, past and current diagnosis and treatment categories). If you filled out the behavioural & social sciences study design questions and have nothing to add here, write "See above."* |
| Recruitment | *Describe how participants were recruited. Outline any potential self-selection bias or other biases that may be present and how these are likely to impact results.* |
| Ethics oversight | Human sera samples were obtained from Seattle Children's Hospital during routine blood draws from children receiving medical care in December 2023. This was approved by the Seattle Children's Hospital Institutional Review Board with a waiver of consent. |

Note that full information on the approval of the study protocol must also be provided in the manuscript.

# Field-specific reporting

Please select the one below that is the best fit for your research. If you are not sure, read the appropriate sections before making your selection.

☒ Life sciences          ☐ Behavioural & social sciences          ☐ Ecological, evolutionary & environmental sciences

For a reference copy of the document with all sections, see nature.com/documents/nr-reporting-summary-flat.pdf

# Life sciences study design

All studies must disclose on these points even when the disclosure is negative.

| | |
|---|---|
| Sample size | 4 randomly selected children sera were used to perform serum escape deep mutational scanning experiments. This number was sufficient to capture heterogeneous individual escape profiles that explain evidence of pleiotropy and epistasis during recent H3N2 evolution, though we do not claim this number is fully representative of the population. |
| Data exclusions | No data was excluded. |
| Replication | Deep mutational scanning experiments were performed using two biological replicates (two independently generated libraries). Validation experiments were performed in biological or technical duplicate as indicated in figure legends. |
| Randomization | No randomization was necessary since we do not make any comparisons between experimental groups. |

| Blinding | No blinding was necessary since we do not make any comparisons between experimental groups and this is not a trial study. |

# Reporting for specific materials, systems and methods

We require information from authors about some types of materials, experimental systems and methods used in many studies. Here, indicate whether each material, system or method listed is relevant to your study. If you are not sure if a list item applies to your research, read the appropriate section before selecting a response.

## Materials & experimental systems

| n/a | Involved in the study |
|---|---|
| ☒ | ☐ Antibodies |
| ☐ | ☒ Eukaryotic cell lines |
| ☒ | ☐ Palaeontology and archaeology |
| ☒ | ☐ Animals and other organisms |
| ☒ | ☐ Clinical data |
| ☒ | ☐ Dual use research of concern |
| ☒ | ☐ Plants |

## Methods

| n/a | Involved in the study |
|---|---|
| ☒ | ☐ ChIP-seq |
| ☒ | ☐ Flow cytometry |
| ☒ | ☐ MRI-based neuroimaging |

## Eukaryotic cell lines

Policy information about cell lines and Sex and Gender in Research

| Cell line source(s) | The following cell lines were used: 293T (ATCC, CRL-3216), 293T-rtTA (from ref 23), 293T-CMV-PB1 (from ref 70), MDCK-SIAT1 (HPA Cultures, 05071502), MDCK-SIAT1-CMV-PB1 (from ref 70), and MDCK-SIAT1-CMV-PB1-TMPRSS2 (from ref 19). |
|---|---|
| Authentication | None were authenticated. |
| Mycoplasma contamination | Not contaminated. |
| Commonly misidentified lines (See ICLAC register) | N/A |

## Plants

| Seed stocks | *Report on the source of all seed stocks or other plant material used. If applicable, state the seed stock centre and catalogue number. If plant specimens were collected from the field, describe the collection location, date and sampling procedures.* |
|---|---|
| Novel plant genotypes | *Describe the methods by which all novel plant genotypes were produced. This includes those generated by transgenic approaches, gene editing, chemical/radiation-based mutagenesis and hybridization. For transgenic lines, describe the transformation method, the number of independent lines analyzed and the generation upon which experiments were performed. For gene-edited lines, describe the editor used, the endogenous sequence targeted for editing, the targeting guide RNA sequence (if applicable) and how the editor was applied.* |
| Authentication | *Describe any authentication procedures for each seed stock used or novel genotype generated. Describe any experiments used to assess the effect of a mutation and, where applicable, how potential secondary effects (e.g. second site T-DNA insertions, mosiacism, off-target gene editing) were examined.* |

