## [Peer Review File · Nature Ecology & Evolution]

Pleiotropic mutational effects on function and stability constrain the antigenic evolution of influenza hemagglutinin

Corresponding Author: Professor Jesse Bloom

Version 0:

Decision Letter:

10th July 2025

Dear Professor Bloom,

Your manuscript entitled "Pleiotropic mutational effects on function and stability constrain the antigenic evolution of influenza hemagglutinin" has now been seen by three reviewers, whose comments are attached. The reviewers have raised a number of concerns which will need to be addressed before we can offer publication in Nature Ecology & Evolution. We will therefore need to see your responses to the criticisms raised and to some editorial concerns, along with a revised manuscript, before we can reach a final decision regarding publication.

We therefore invite you to revise your manuscript taking into account all reviewer and editor comments. Please highlight all changes in the manuscript text file.

* If you have not done so already please begin to revise your manuscript so that it conforms to our Article format instructions at <http://www.nature.com/natecolevol/info/final-submission>. Refer also to any guidelines provided in this letter.

* Extended Data Figures - please ensure that any supplementary figures and tables that are crucial to the manuscript's conclusions are converted into Extended Data figures and tables to increase visibility of these data. Extended Data figures and tables are online-only (present in the online PDF and full-text HTML versions of the paper), peer-reviewed display items that provide essential background to the article but are not included in the main article due to space constraints. A maximum of ten Extended Data display items (figures and tables) is permitted.

Link Redacted

Nature Ecology & Evolution is committed to improving transparency in authorship. As part of our efforts in this direction, we are now requesting that all authors identified as 'corresponding author' on published papers create and link their Open Researcher

and Contributor Identifier (ORCID) with their account on the Manuscript Tracking System (MTS), prior to acceptance. ORCID helps the scientific community achieve unambiguous attribution of all scholarly contributions. You can create and link your ORCID from the home page of the MTS by clicking on 'Modify my Springer Nature account'. For more information please visit www.springernature.com/orcid.

[redacted]

Reviewer expertise:

Reviewer #1: viral antigenic evolution

Reviewer #2: RNA viruses, evolutionary dynamics, mathematical models

Reviewer #3: RNA viruses, viral evolution, experimental virology

Reviewer comments:

Reviewer #1 (Remarks to the Author):

In this manuscript, Yu and colleagues use deep mutation scanning (DMS) to define the effects of mutations in the hemagglutinin (HA) protein of a recent strain of the H3N2 influenza virus on viral entry and escape from antibodies. After generating a DMS library of the H3 protein, they use a lentiviral-based DMS platform to define the effects of amino acid substitutions on cell entry, and complete additional experiments using pH treatment of lentiviral libraries to quantify impacts of HA mutations on pH stability. Using these DMS experiments, the authors find that, consistent with previous reports, some substitutions within the HA receptor-binding site (RBS) are often subject to epistatic entrenchment, where reversion to the ancestral sequence results in considerable fitness costs. In contrast, with one exception mutations observed outside the RBS did not exhibit epistatic entrenchment. The authors demonstrate that the fitness costs of a recent antigenic substitution (I140K) were likely offset by other mutations, indicating epistatic interactions enabled this antigenically beneficial mutation to fix. Finally, the authors confirm the effects of many of these substitutions using a conditionally replicative influenza system, allowing them to extend their findings to authentic influenza virus and minimize potential artifacts of the lentiviral based DMS platform.

The manuscript is well-written and the figures are logical and easy to follow. In addition, the conclusions drawn by the authors are well supported by the data. There are a couple of areas where the authors could expand a bit on their findings, which would improve the manuscript:

Major points:

1. The authors nicely dissect how mutations to site 165 or 167 destabilize the HA, likely because they result in the loss of an N-linked glycan at position 165 that may stabilize protomers. The T248N substitution for which the authors observe a loss of binding as well as evidence for epistatic entrenchment would also be predicted to result in the loss of an N-linked glycan at 246, and in addition it looks like T246S has a markedly less deleterious impact on cell entry than other substitutions at the same position. The authors might consider commenting on whether they think the presence of this glycan might be related to this observed entrenchment, and what the potential role might be (pH stability, receptor binding, etc).

2. Some mAbs have been described that bind across HA protomers (PMID: 24719430) – is it possible that some of the substitutions which alter pH stability and also strongly escape sera are escaping antibody specificities that target quaternary epitopes?

3. The authors note recent work that demonstrates a structural basis for the G186D and D190N epistatic entrenchment that they also find in their data – is there a similar structural explanation for the Y195F entrenchment observed by the authors? In other words, do the authors have some idea of what potential epistatic partners could be constraining the reversion to Y195?

Reviewer #1 (Remarks on code availability):

Although I lack the technical expertise to review the code for analysis the code is deposited, publicly available, and easily found.

Reviewer #2 (Remarks to the Author):

In this manuscript, Yu and coauthors use pseudovirus deep mutational scanning (DMS) of influenza's hemagglutinin (HA) protein to consider the impact that epistasis and pleiotropy have on HA antigenic evolution. They find that many substitutions that impact antigenicity (both in the RBP and in the characterized epitopes A-E) also impact cell entry, with most substitutions having a deleterious effect (Figure 1). They also find that most substitutions that impact HA acid stability are deleterious (Figure 2). Through a DMS analysis of the H3N2 MA22 strain, the authors also show that there is evidence for epistatic entrenchment, with genetic changes in the F195 lineage reducing the ability of F to revert back to Y (Figure 3). Finally, the authors show in Figure 4 an example of permissive mutations allowing for the emergence of an antigenic mutation that was not able to spread

beforehand due to its impact on cell entry. Overall, this paper presents interesting analyses that highlight the role that pleiotropy plays in modulating antigenic evolution and documenting epistatic interactions (both entrenchment and permissive mutations) in the context of antigenic changes in the HA.

Major comments:

Given considerable amounts of previous work on epistasis in influenza virus proteins (including analyses that have documented entrenchment and permissive mutations, as cited in the manuscript), I think the manuscript text needs to more strongly elaborate on the novelty of its findings, particularly what new understanding is gained from this specific study.

The text does not go into detail about what evolutionary processes drive the substitutions that cause entrenchment. Is genetic drift responsible for entrenchment-causing substitutions Y159N and T160I in the derived F195 lineage? Or is it positive selection that is driving these substitutions in the F195 background? Furthermore, couldn't Y195F be deleterious with respect to its impact on cell entry/function but have a net beneficial effect with respect to population-level spread if there is sufficient immune escape that this substitution confers? If there is a cell entry fitness cost to an immune escape mutation, are substitutions Y159N and T160I compensatory mutations that also just happen to cause entrenchment?

Re. the last sentence of the abstract: 'Our results refine our understanding of the mutational constraints that shape influenza evolution: epistasis can enable antigenic change, but pleiotropic effects can restrict its trajectory.' This seems to me to be a broad conclusion based on experimental results from a very small number of strains. Although the work is rigorously done, perhaps add a sentence to indicate that future work needs to determine the extent to which this conclusion holds across other H3N2 strains and across its evolutionary trajectory?

I'm curious about the generality of the results that pleiotropic effects restrict the antigenic evolution of influenza. Does this conclusion depend on how well-adapted an influenza virus is to its host? In a less well-adapted virus, could more cell entry/function mutations be beneficial, such that pleiotropic effects would not constrain antigenic evolution to the same extent?

Considerable amounts of text and the entirety of Figure 3 is focused on entrenchment. However, entrenchment is not mentioned in the abstract and its role in HA "forward evolution" is not discussed. Why is the documentation of entrenchment important for understanding constraints on HA antigenic evolution?

p.4 and Figure 1: It seems that two factors impact the entropy observed in natural sequences: the extent of immune pressure at a site as well as the extent of cell entry constraints (e.g., Figure 1E: epitope B has high entropy despite having strong constraints relative to the other 4 epitopes, with the text saying this is likely because of this epitope's immunodominance; Figure 1D: outside RBP has low entropy despite similar constraints to 190-helix and 220-loop presumably because of only low immune pressure on sites outside the RBP). This might be worthwhile to point out explicitly in the text, and at some point mention that both of these factors can change over time (immune pressure with changes in population-level immunity, and constraints with changes in genetic background)?

Figure 1D & E: entropy: how do the authors think entropy measurements relate to the extent that positive selection has occurred in a region/epitope? Has it been demonstrated that entropy faithfully measures the extent to which selection can act on a region in the viral genome? Also, more information on how entropy was calculated from H3N2 sequences since 1968 would be helpful. Were all available HA sequences from Genbank used? Was there downsampling of some sort to account for more deposited sequences in more recent times?

Minor:

Please add line numbers in a revision

Reviewer #3 (Remarks to the Author):

Yu and Bloom, et al. present a deep mutational scanning study of HA of Influenza, exploring the pleiotropic effects of mutations on binding/entry, stability and antigen escape. The study is well done, and the paper well-written. It provides a rare example of DMS studies that explore multiple phenotypes simultaneously to explore the pleiotropic effects of mutations on proteins, and the role of epistatic entrenchment in influencing important evolutionary outcomes.

In my opinion, one weakness with the manuscript is in the acid stability portion, where fitness is examined across a range of pH. Although this provides a rich data set to examine the titration of fitness across this gradient, the authors do little to examine how specific substitutions respond. In general the authors rely on mean fitness of substitution, and neglect to dig more deeply into the specific substitutions that respond to pH change in HA. Understanding these constraints could be very interesting and is worth some effort.

With so much missing data (due to the masking of acid stability by effects on binding/entry), it's unclear how to interpret the structural representations in Figure 2, and how meaningful they are. Since binding is dependent on stability, can substitutions be assessed for correct folding, and distinguished from those that are unstable even at neutral pH? This might be possible with a conformational antibody, sorting and sequencing of the folded HA presented on the surface of cells?

The analysis presented in Figure 3 is especially interesting, but I felt the discussion of potential explanations for the occurrence of entrenchment, however given the modest effects on acid stability shown in S2D, is this result surprising or important? More likely those mutations with strong effects on stability are removed from the analysis due to their effects on binding/entry?

Untangling the independence of those phenotypes is especially important to understanding the importance and significance of the findings in this manuscript. The discussion provides explanations of why epistatic entrenchment may be more common in cell entry than in acid stability, however these feel somewhat superficial.

Reviewer #3 (Remarks on code availability):

The code is well organized and provides adequate usage information.

*****END*****

Version 1:

Decision Letter:

4th September 2025

Dear Dr. Bloom,

Thank you for submitting your revised manuscript "Pleiotropic mutational effects on function and stability constrain the antigenic evolution of influenza hemagglutinin" (NATECOLEVOL-25051671A). It has now been seen again by the original reviewers and their comments are below. The reviewers find that the paper has improved in revision, and therefore we'll be happy in principle to publish it in Nature Ecology & Evolution, pending minor revisions to comply with our editorial and formatting guidelines.

If you have not done so already, please ensure that you also email us a completed copy of the Reporting summary :

Reporting summary: <https://www.nature.com/documents/nr-reporting-summary.pdf>

[redacted]

Reviewer #1 (Remarks to the Author):

The authors have adequately addressed my comments and those of other reviewers and I have no further concerns.

Reviewer #2 (Remarks to the Author):

I have reviewed the revised manuscript and the authors have satisfactorily addressed all of my initial concerns.

Reviewer #3 (Remarks to the Author):

I appreciate the detailed responses to my comments. The additional analysis is helpful (of course there is much more that could be dug into here) and the additions to text address many of my concerns. This is a very nice addition to the literature, and I am supportive of publication.

Version 2:

Decision Letter:

8th October 2025

Dear Professor Bloom,

We are pleased to inform you that your Article entitled "Pleiotropic mutational effects on function and stability constrain the antigenic evolution of influenza hemagglutinin", has now been accepted for publication in Nature Ecology & Evolution.

Over the next few weeks, your paper will be copyedited to ensure that it conforms to Nature Ecology and Evolution style. Once your paper is typeset, you will receive an email with a link to choose the appropriate publishing options for your paper and our Author Services team will be in touch regarding any additional information that may be required

Due to the importance of these deadlines, we ask you please us know now whether you will be difficult to contact over the next month. If this is the case, we ask you provide us with the contact information (email, phone and fax) of someone who will be able to check the proofs on your behalf, and who will be available to address any last-minute problems. Once your paper has been scheduled for online publication, the Nature press office will be in touch to confirm the details.

Acceptance of your manuscript is conditional on all authors' agreement with our publication policies (see www.nature.com/authors/policies/index.html). In particular your manuscript must not be published elsewhere and there must be no announcement of the work to any media outlet until the publication date (the day on which it is uploaded onto our web site).

Authors may need to take specific actions to achieve compliance with funder and institutional open access mandates. If your research is supported by a funder that requires immediate open access (e.g. according to [Plan S principles](https://www.springernature.com/gp/open-science/plan-s-compliance) or the [NIH public access policy](https://www.springernature.com/gp/open-science/us-federal-agency-compliance)) then you should select the gold OA route, and we will direct you to the compliant route where possible. Because authors warrant under our subscription licensing terms that they haven't committed to licensing any version of their article under a licence inconsistent with the terms of our agreement – including the applicable embargo period – publication under the subscription model isn't suitable for authors whose funders require no embargo.

We welcome the submission of potential cover material (including a short caption of around 40 words) related to your manuscript; suggestions should be sent to Nature Ecology & Evolution as electronic files (the image should be 300 dpi at 210 x 297 mm in either TIFF or JPEG format). Please note that such pictures should be selected more for their aesthetic appeal than for their scientific content, and that colour images work better than black and white or grayscale images. Please do not try to design a cover with the Nature Ecology & Evolution logo etc., and please do not submit composites of images related to your work. I am sure you will understand that we cannot make any promise as to whether any of your suggestions might be selected for the cover of the journal.

You can generate the link yourself when you receive your article DOI by entering it here: <http://authors.springernature.com/share>.

Thank you for choosing NEE to publish your work. I look forward to seeing it published soon.

[redacted]

P.S. Click on the following link if you would like to recommend Nature Ecology & Evolution to your librarian <http://www.nature.com/subscriptions/recommend.html#forms>

** Visit the Springer Nature Editorial and Publishing website at http://editorial-jobs.springernature.com?utm_source=ejP_NEcoE_email&utm_medium=ejP_NEcoE_email&utm_campaign=ejp_NEcoE for more information about our career opportunities. If you have any questions please click [here](mailto:editorial.publishing.jobs@springernature.com).

Reviewer #1 (Remarks to the Author):

In this manuscript, Yu and colleagues use deep mutation scanning (DMS) to define the effects of mutations in the hemagglutinin (HA) protein of a recent strain of the H3N2 influenza virus on viral entry and escape from antibodies. After generating a DMS library of the H3 protein, they use a lentiviral-based DMS platform to define the effects of amino acid substitutions on cell entry, and complete additional experiments using pH treatment of lentiviral libraries to quantify impacts of HA mutations on pH stability. Using these DMS experiments, the authors find that, consistent with previous reports, some substitutions within the HA receptor-binding site (RBS) are often subject to epistatic entrenchment, where reversion to the ancestral sequence results in considerable fitness costs. In contrast, with one exception mutations observed outside the RBS did not exhibit epistatic entrenchment. The authors demonstrate that the fitness costs of a recent antigenic substitution (I140K) were likely offset by other mutations, indicating epistatic interactions enabled this antigenically beneficial mutation to fix. Finally, the authors confirm the effects of many of these substitutions using a conditionally replicative influenza system, allowing them to extend their findings to authentic influenza virus and minimize potential artifacts of the lentiviral based DMS platform.

The manuscript is well-written and the figures are logical and easy to follow. In addition, the conclusions drawn by the authors are well supported by the data. There are a couple of areas where the authors could expand a bit on their findings, which would improve the manuscript:

Thanks for the positive summary. We are glad the reviewer thought our manuscript was well written and conclusions well supported.

Major points:

1. The authors nicely dissect how mutations to site 165 or 167 destabilize the HA, likely because they result in the loss of an N-linked glycan at position 165 that may stabilize protomers. The T248N substitution for which the authors observe a loss of binding as well as evidence for epistatic entrenchment would also be predicted to result in the loss of an N-linked glycan at 246, and in addition it looks like T246S has a markedly less deleterious impact on cell entry than other substitutions at the same position. The authors might consider commenting on whether they think the presence of this glycan might be related to this observed entrenchment, and what the potential role might be (pH stability, receptor binding, etc).

We thank the reviewer for bringing this to our attention. We assume the reviewer is referring to T248S (not T246S), which is the sole mutation that preserves the N-linked glycan at N246. We agree it is interesting that T248S is better tolerated than other mutations at sites 246/248. Since the N246 glycan was acquired in the 1980s, it appears to have been non-essential in the past (pre-1980s) but has been conserved since then.

Prior work has shown that glycosylation sites near the receptor binding pocket shield epitopes from antibodies, but often impose a fitness cost (Das et al. 2011). Site 246 is located next to the receptor binding pocket (~13Å from sialic acid). Therefore, we speculate that antigenic pressure drove the N246 glycosylation site via an N248T mutation to fixation in the 1980s. Mutations in the region that either compensated for the glycan or were contingent on the glycan led to its entrenchment, explaining why removing the glycan is now highly deleterious to cell entry in the recent MA22 background.

We have incorporated these details into the following section:

Line 238: In contrast to the extensive epistatic entrenchment involving mutations in the receptor-binding pocket with respect to cell entry, we saw little evidence of entrenchment with respect to cell entry involving mutations in other regions of HA. Nearly all reversions to ancestral amino acids at sites outside of the receptor binding pocket are well tolerated with respect to cell entry with the single exception of T248N (**Fig. 3B**). Interestingly, the N248T mutation that fixed in the 1980s created an N-linked glycosylation motif at N246 that has been maintained ever since. T248S (which is the only mutation at site 248 that retains this motif) is noticeably more tolerated than other mutations at sites 246 and 248 with respect to cell entry (**Extended Data Fig. 3**), indicating the glycan is now entrenched. In H3N2 HA evolution, the acquisition of glycosylation sites near the receptor binding pocket can shield epitopes from antibodies, but often imposes a fitness cost⁵⁰. Therefore, the N248T mutation—which is located near the receptor binding pocket—was likely selected by antigenic pressure, and mutations in the region that compensated for or became dependent on the glycan led to its entrenchment.

2. Some mAbs have been described that bind across HA protomers (PMID: 24719430) – is it possible that some of the substitutions which alter pH stability and also strongly escape sera are escaping antibody specificities that target quaternary epitopes?

This is an interesting point and we thank the reviewer for bringing this citation to our attention. We agree antibodies that bind across protomers to prevent pH-triggered HA destabilization could be a possible explanation for our data. We have modified our discussion and included the citation:

Line 380: While it is unclear how these mutations reduce serum antibody neutralization, we speculate there are two possible mechanisms. These mutations could abrogate **binding of antibodies that directly target the trimer interface, nearby epitopes, or across protomers**⁶⁵. Alternatively, the destabilizing effect could accelerate membrane fusion, a previously reported strategy for antibody escape⁶⁶.

3. The authors note recent work that demonstrates a structural basis for the G186D and D190N epistatic entrenchment that they also find in their data – is there a similar structural explanation for the Y195F entrenchment observed by the authors? In other words, do the authors have some idea of what potential epistatic partners could be constraining the reversion to Y195?

Yes, N159 and I160 constrain the reversion back to Y195. Reviewer 2 was also interested in this epistatic example and suggested we discuss it in more detail. See our response to the Reviewer 2 comment on this point for an explanation of how we revised our manuscript to discuss this topic.

Reviewer #1 (Remarks on code availability):

Although I lack the technical expertise to review the code for analysis the code is deposited, publicly available, and easily found.

We are happy to hear the code was easy to access.

Reviewer #2 (Remarks to the Author):

In this manuscript, Yu and coauthors use pseudovirus deep mutational scanning (DMS) of influenza's hemagglutinin (HA) protein to consider the impact that epistasis and pleiotropy have on HA antigenic evolution. They find that many substitutions that impact antigenicity (both in the RBP and in the characterized epitopes A-E) also impact cell entry, with most substitutions having a deleterious effect (Figure 1). They also find that most substitutions that impact HA acid stability are deleterious (Figure 2). Through a DMS analysis of the H3N2 MA22 strain, the authors also show that there is evidence for epistatic entrenchment, with genetic changes in the F195 lineage reducing the ability of F to revert back to Y (Figure 3). Finally, the authors show in Figure 4 an example of permissive mutations allowing for the emergence of an antigenic mutation that was not able to spread beforehand due to its impact on cell entry. Overall, this

paper presents interesting analyses that highlight the role that pleiotropy plays in modulating antigenic evolution and documenting epistatic interactions (both entrenchment and permissive mutations) in the context of antigenic changes in the HA.

Thanks for the nice summary of our work. We are glad to hear the reviewer found our analyses to be interesting.

Major comments:

Given considerable amounts of previous work on epistasis in influenza virus proteins (including analyses that have documented entrenchment and permissive mutations, as cited in the manuscript), I think the manuscript text needs to more strongly elaborate on the novelty of its findings, particularly what new understanding is gained from this specific study.

We thank the reviewer for this suggestion and agree we can describe the novelty of our findings more. A major advance of our work is the application of deep mutational scanning to measure mutation effects on additional phenotypes, in particular acid stability. Doing so enabled us to explore how pleiotropic effects of mutations have constrained evolution, a study that hasn't been performed before. We have re-written the beginning of our Discussion to emphasize this point more.

Line 345: The extent to which pleiotropic conflicts constrain evolution of H3N2 HA has remained unclear. Answering this question requires probing how mutations affect multiple distinct HA molecular phenotypes. Here, we measured the effects of all amino acid mutations to a recent H3 HA on cell entry, acid stability, and neutralization by serum antibodies.

We agree there is considerable amount of work documenting entrenchment of mutations within the receptor binding pocket, but to our knowledge, no study has investigated whether similar entrenchment is observed outside the receptor binding pocket or with respect to a different molecular phenotype that does not depend on receptor binding. We explore both in our work, and have modified the following Discussion section to clarify this novelty.

Line 353: Other studies have reported epistatic entrenchment to be common with respect to the impact of HA mutations within the receptor binding pocket on viral replication^{9-12,14}. Our work also finds that HA mutations in the receptor binding pocket have become entrenched with respect to their effects on cell entry, but this pattern is observed much less often for HA mutations outside of the receptor binding pocket. Furthermore, epistatic entrenchment with respect to acid stability is absent across the entire HA during the timeframe we analyzed, demonstrating how epistasis can be phenotype-specific.

The text does not go into detail about what evolutionary processes drive the substitutions that cause entrenchment. Is genetic drift responsible for entrenchment-causing substitutions Y159N

and T160I in the derived F195 lineage? Or is it positive selection that is driving these substitutions in the F195 background? Furthermore, couldn't Y195F be deleterious with respect to its impact on cell entry/function but have a net beneficial effect with respect to population-level spread if there is sufficient immune escape that this substitution confers? If there is a cell entry fitness cost to an immune escape mutation, are substitutions Y159N and T160I compensatory mutations that also just happen to cause entrenchment?

Reviewer #1 was also interested in this point and we agree more discussion would be helpful. The effects of Y195F, Y159N, and T160I on receptor binding were recently characterized in Liang et al. 2025. Individually, Y195F and T160I have neutral effects on receptor binding. Y159N completely abrogates receptor binding and cannot be rescued by T160I alone. However, the Y159N/T160I/Y195F triple mutant not only restores receptor binding, but also expands glycan binding breadth beyond that of the Y195F single mutant.

To explore the evolutionary process, we used data from a sub-sampled Nextstrain tree to compute the frequencies of all combinations of amino acids observed at sites 159, 160, and 195 since 2018. Haplotypes that reach a frequency >20% at some timepoint are colored according to the key, while other haplotypes are colored gray. Y159N and T160I only arise to fixation in the background of 195F, and this triple mutant lineage (yellow) eventually outcompetes the lineage containing 195F alone (light green). This is consistent with both Liang et al. 2025 and our work—Y195F was a permissive mutation that enabled Y159N and T160I to fix. We do not think Y159N and T160I fixed by drift. Instead, these mutations were likely adaptive as they expand receptor binding specificity and enhance antigenicity. Y195F, Y159N, and T160I are all located in antigenic region B, and the latter two mutations have been shown to escape antibodies (Welsh et al. 2024, Bolton et al. 2022). Taken together, the triple mutant likely possessed a fitness advantage that enabled it to outcompete the lineage containing Y195F alone.

We have included this as **Extended Data Fig. 6** and added text to the following section:

Line 215: For instance, a mutation at site 195 from Y to F emerged in 2020 and swept to fixation among human H3N2 strains, but the reversion F195Y in the MA22 background is highly deleterious to cell entry, indicating the mutation of site 195 from Y to F has become entrenched (**Fig. 3B**). *Y195F was recently shown to be a permissive mutation*

that enabled the fixation of Y159N and T160I (**Extended Data Fig. 6**), which confer antigenic benefits and expand receptor specificity in the presence of 195F but impair receptor binding when paired with 195Y^{14,16,49}. As the MA22 HA contains 159N and 160I, the reversion F195Y is no longer accessible because it disrupts receptor binding and subsequent cell entry.

Re. the last sentence of the abstract: 'Our results refine our understanding of the mutational constraints that shape influenza evolution: epistasis can enable antigenic change, but pleiotropic effects can restrict its trajectory.' This seems to me to be a broad conclusion based on experimental results from a very small number of strains. Although the work is rigorously done, perhaps add a sentence to indicate that future work needs to determine the extent to which this conclusion holds across other H3N2 strains and across its evolutionary trajectory?

We agree that statement was too broad. We have toned down our abstract to clarify that our study specifically studies recent H3N2 evolution:

Line 33: Our results refine our understanding of the mutational constraints that shape **recent H3N2** influenza evolution: epistasis can enable antigenic change, but pleiotropic effects can restrict its trajectory.

Additionally, we have incorporated the following in the discussion:

Line 389: The extent to which pleiotropic constraints can be alleviated by epistasis differs across phenotypes: for example, constraints on cell entry appear more readily alleviated than constraints on acid stability **in recent H3N2 influenza HA evolution. Whether these constraints similarly shape the evolution of other H3N2 strains and influenza subtypes remains to be determined.**

I'm curious about the generality of the results that pleiotropic effects restrict the antigenic evolution of influenza. Does this conclusion depend on how well-adapted an influenza virus is to its host? In a less well-adapted virus, could more cell entry/function mutations be beneficial, such that pleiotropic effects would not constrain antigenic evolution to the same extent?

Yes, we think this is likely the case. There is evidence for SARS-CoV-2 that early adaptation of a novel virus to humans can involve single mutations that are strongly adaptive to transmission in the new host (e.g., D614G or N501Y for SARS-CoV-2 spike), but over time these "easy" adaptations are used up. However, since our study characterizes a recent human H3N2 HA that is well-adapted to humans, we believe it is out of the scope of this study to evaluate if there exist fewer pleiotropic constraints in less well-adapted H3N2 strains. In addition, there is insufficient sequence sampling of H3N2 in the first half-decade after it entered humans (1968-1973) to track its evolution in detail, as only a small number of mostly somewhat egg- or cell-adapted strains from that timeframe are available.

Considerable amounts of text and the entirety of Figure 3 is focused on entrenchment. However, entrenchment is not mentioned in the abstract and its role in HA “forward evolution” is not discussed. Why is the documentation of entrenchment important for understanding constraints on HA antigenic evolution?

We thank the reviewer for this suggestion. We originally chose to omit the term “entrenchment” in the abstract out of concern that its meaning might be unclear. This is why we devoted a paragraph in the main text (**Line 206**) and a main figure panel (**Fig. 3A**) to explaining the concept of entrenchment. However, we agree that entrenchment should be mentioned in the abstract given the substantial amount of text that is dedicated to it. We’ve added the following in the abstract:

Line 28: We find that epistasis has **entrenched certain mutations so that reverting to the ancestral amino acid identity in earlier strains is no longer tolerated. Epistasis has also enabled** the emergence of antigenic mutations that were detrimental to HA’s cell entry function in earlier strains.

We have also added more discussion of how entrenchment influences forward evolution of HA, and why studying entrenchment is important for understanding evolutionary constraints.

Line 230: Collectively, these results along with prior work^{9–12,14}, highlight the existence of extensive epistasis with respect to the effects of mutations within the HA receptor binding pocket on cell entry. **This epistasis restricts access to ancestral amino acids and simultaneously opens new evolutionary paths for antigenic change (e.g., 159N, G186D). However, not all mutations become entrenched, since reversions to ancestral amino acids are observed in HA evolution. Therefore, analysis of entrenchment allows us to determine which reversions are currently accessible or constrained.**

p.4 and Figure 1: It seems that two factors impact the entropy observed in natural sequences: the extent of immune pressure at a site as well as the extent of cell entry constraints (e.g., Figure 1E: epitope B has high entropy despite having strong constraints relative to the other 4 epitopes, with the text saying this is likely because of this epitope’s immunodominance; Figure 1D: outside RBP has low entropy despite similar constraints to 190-helix and 220-loop presumably because of only low immune pressure on sites outside the RBP). This might be worthwhile to point out explicitly in the text, and at some point mention that both of these factors can change over time (immune pressure with changes in population-level immunity, and constraints with changes in genetic background)?

We thank the reviewer for this suggestion and have incorporated this point into the following section:

Line 136: This discrepancy likely reflects the immunodominance of epitope B with respect to antibody neutralization in the human population, which imposes strong

positive selection for mutations at sites within the region^{33–35}. Therefore, variability observed in natural sequences depends on both the extent of immune pressure at a site as well as constraints on HA function. Both constraints can change over time as a result of shifts in population-level immunity and HA genetic background.

Figure 1D & E: entropy: how do the authors think entropy measurements relate to the extent that positive selection has occurred in a region/epitope? Has it been demonstrated that entropy faithfully measures the extent to which selection can act on a region in the viral genome? Also, more information on how entropy was calculated from H3N2 sequences since 1968 would be helpful. Were all available HA sequences from Genbank used? Was there downsampling of some sort to account for more deposited sequences in more recent times?

We used entropy as we are really trying to assess the variability of a site in natural evolution rather than the strength of positive selection per se. In any case, we think entropy is a reasonable proxy for the extent of positive selection at a site in a tree of H3N2 influenza that is subsampled evenly across time like the tree we used. First, positions that have undergone multiple amino acid sweeps tend to have higher entropy values. For example, site 144 has the highest entropy (1.601), is located in a known antigenic site, and is certainly under positive selection due to population immunity.

To more rigorously show this, we estimated a site-specific dN/dS using a fixed effects likelihood (FEL) with HyPhy. The same subsampled Nextstrain tree that the entropy values were derived from was used as input. We calculated the median dN/dS across sites in each receptor binding pocket or antigenic region, and the result matches closely with the mean entropy values across sites in these regions shown in **Fig. 1D** and **1E**. Therefore, this further indicates that entropy is accurately capturing the extent of positive selection at sites.

We have included information on how sequences were downsampled in the Nextstrain tree and how entropy was calculated to the Methods under section “Entropy calculation from natural sequences”.

Minor:

Please add line numbers in a revision

Line numbers have been added in the revision.

Reviewer #3 (Remarks to the Author):

Yu and Bloom, et al. present a deep mutational scanning study of HA of Influenza, exploring the pleiotropic effects of mutations on binding/entry, stability and antigen escape. The study is well done, and the paper well-written. It provides a rare example of DMS studies that explore multiple phenotypes simultaneously to explore the pleiotropic effects of mutations on proteins, and the role of epistatic entrenchment in influencing important evolutionary outcomes.

Thanks for the positive summary of our work. We are glad the reviewer thought the study was well done and well-written.

In my opinion, one weakness with the manuscript is in the acid stability portion, where fitness is examined across a range of pH. Although this provides a rich data set to examine the titration of fitness across this gradient, the authors do little to examine how specific substitutions respond. In general the authors rely on mean fitness of substitution, and neglect to dig more deeply into

the specific substitutions that respond to pH change in HA. Understanding these constraints could be very interesting and is worth some effort.

We thank the reviewer for this suggestion. In our manuscript, we discuss the effects of specific mutations on acid stability in two contexts: the N-linked glycosylation at sites 165/167 (**Fig. 2E**) and a tetrad salt bridge (**Fig. 2G**). We agree deeper exploration of how specific mutations affect acid stability in HA beyond these few cases would be helpful.

In particular, we took a closer look at sites 219 and 223, which harbor many destabilizing mutations yet have changed during H3N2 evolution (**Extended Data Fig. 7B**). These sites are located within or near the receptor binding pocket and antigenic regions, providing a relevant opportunity to study how specific mutations constrain evolution. Furthermore, many mutations at these sites are tolerated with respect to cell entry. However, while many mutations are tolerated at neutral pH, some start to have a destabilizing effect at lower pH. At sites 219, this is the case for the charged amino acids D, K, and R. At sites 223, this is the case for all non-hydrophobic mutations and A. These destabilizing mutations are never selected by evolution.

We have included this as **Extended Data Fig. 7D** and added the following text:

Line 258: At the destabilizing sites that do show variation among natural sequences, the natural mutations are exclusively the particular amino-acid changes that do not affect stability (**Extended Data Fig. 7B**). For example, most mutations at site 219 and 223 do not impair cell entry, but some destabilize HA at acidic pH (**Extended Data Fig. 7D**). At site 219, the aromatic residues tyrosine and phenylalanine have appeared in natural strains, although neither fixed (**Extended Data Fig. 7C**), whereas charged residues (aspartic acid, lysine, and arginine) have a destabilizing effect (**Extended Data Fig. 7D**) and are never observed in natural strains. Similarly, site 223 has exchanged bulky hydrophobic residues valine and isoleucine during natural evolution of H3N2 HA, and only large hydrophobic amino acids at this site preserve acid stability (**Extended Data Fig. 7C**). By contrast, the many other amino-acid identities at site 223 are destabilizing (**Extended Data Fig. 7D**) and not observed in natural strains.

We hope these examples demonstrate how specific substitutions may be constrained during evolution. While it isn't possible to explore these constraints at every destabilizing site in the paper, our dataset should enable others to analyze the effects of specific mutations in greater detail with formal molecular modeling. This was part of the motivation for creating an interactive webpage that makes the data easy to access (see https://dms-vep.org/Flu_H3_Massachusetts2022_DMS/acid_stability.html), so our data can be leveraged for continued study of how specific mutations affect stability, including for purposes like basic biochemical studies and perhaps HA stabilization for vaccines.

With so much missing data (due to the masking of acid stability by effects on binding/entry), it's unclear how to interpret the structural representations in Figure 2, and how meaningful they are. Since binding is dependent on stability, can substitutions be assessed for correct folding, and distinguished from those that are unstable even at neutral pH? This might be possible with a conformational antibody, sorting and sequencing of the folded HA presented on the surface of cells?

We thank the reviewer for bringing up this concern. We agree the structural representations in the original manuscript were unclear, as a site that is colored white could correspond to a site where 1) mutations genuinely do not affect acid stability or 2) mutations disrupt HA folding or function so badly that it isn't possible to measure the effect of a mutation on acid stability.

To clarify this, we colored sites that are missing all acid stability measurements dark gray in the structures shown in Figure 2 (example above). Dark gray sites therefore represent sites where only a single residue is tolerated and all mutations disrupt cell entry (most likely via disrupting folding or function). We note this is imperfect, as some sites that are not colored dark gray may contain just a few residues that are tolerated, while most mutations are deleterious to cell entry. This is a limitation of the structural representations, but a reader interested in this fine detail could refer to **Extended Data Fig. 4** or https://dms-vep.org/Flu_H3_Massachusetts2022_DMS/acid_stability.html for the heatmap that shows which specific mutations were not tolerated for cell entry.

We thank the reviewer for the experimental suggestion, but we think it is out of the scope of this work. Most mutations outside of the receptor binding pocket and the fusion peptide that lead to poor cell entry likely disrupt protein folding at neutral pH, so the results of such an experiment would just strongly correlate with the cell entry effects. In addition, our pseudovirus libraries allow easy measurement of HA function, but not staining for folded protein.

The analysis presented in Figure 3 is especially interesting, but I felt the discussion of potential explanations for the occurrence of entrenchment, however given the modest effects on acid stability shown in S2D, is this result surprising or important? More likely those mutations with strong effects on stability are removed from the analysis due to their effects on binding/entry? Untangling the independence of those phenotypes is especially important to understanding the importance and significance of the findings in this manuscript. The discussion provides explanations of why epistatic entrenchment may be more common in cell entry than in acid stability, however these feel somewhat superficial.

We agree it is important to untangle the independence between cell entry and acid stability. We have added a few sentences to clarify how they are distinct.

Line 162: Importantly, the effects of mutations on acid stability are not strongly correlated with the effects of mutations on cell entry, indicating these assays capture distinct molecular phenotypes (**Extended Data Fig. 2D**). **The phenotypes are distinct for several reasons. First, comparison of natural influenza strains shows that HAs can have acid stabilities that span an appreciable range (human seasonal strains tend to have a fusion pH of 5.0-5.5 whereas avian influenza strains tend to have a fusion pH of 5.6-6.0³⁷⁻³⁹) but still effectively mediate entry in cells in the lab, demonstrating that a range of stabilities are compatible with entry into cell lines even if evolutionary selection for transmissibility in actual human or avian hosts favors a tighter stability range. Second, many mutations that impair cell entry do so by disrupting HA folding, receptor binding, or fusion-mediating conformational changes in a manner that is unrelated to acid stability.**

This independence underlies an important finding in work: some mutations that strongly escape antibodies have no apparent effect on cell entry, but do destabilize HA. These mutations have never fixed in nature, indicating a strong pleiotropic constraint on HA acid stability specifically.

With respect to the reviewers' concerns about **Fig. 3**, we don't think that mutations with strong effects on stability are being removed from the analysis due to effects on entry. In fact, we were able to measure acid stability effects for most mutations, even for many with subpar entry. We encourage the reviewer to explore the visualization at https://dms-vep.org/Flu_H3_Massachusetts2022_DMS/entrenchment.html. If you hover over the red points, very few mutations have a 'null' pH stability effect, which indicates a missing acid stability measurement.

Therefore, we think it is a significant result that mutation effects on acid stability haven't become entrenched in the same way that mutation effects on cell entry have. However, we agree we could do a better job of explaining why this might be the case. We have added to this section to make it less superficial.

Line 359: Why might mutation effects on acid stability be less prone to shift due to epistasis? A variety of studies have found that mutations to proteins often have roughly additive effects on stability^{53,54}, and epistasis at the level of function tends to arise from the non-linear relationship between stability and function rather than underlying epistasis in the effects of mutations on stability^{55,56}. In contrast, the receptor binding pocket involves a network of amino acid residues positioned so that their side chains interact with sialic acid via hydrogen bonds and other non-covalent contacts. Mutations at one site can alter these interactions in ways that modify the effects of changes at interacting sites⁹⁻¹⁴, providing a structural basis for extensive epistasis with respect to cell entry. In a more abstract view, acid stability could represent a single underlying "global" biophysical property that is less influenced by epistasis^{26,57}, whereas cell entry is a higher order phenotype that involves several underlying properties (e.g., receptor-binding, protein stability, and membrane fusion). There is evidence for other proteins that mutations often have additive effects on underlying biophysical properties, and epistasis in higher-order phenotypes often arises simply from their non-linear dependence on underlying molecular properties^{26,58-60}.

Reviewer #3 (Remarks on code availability):

The code is well organized and provides adequate usage information.

We are happy to hear the code is well organized.